# A model study on investigating the sensitivity of aerosol forcing on the volatilities of semi-volatile organic compounds

Muhammed Irfan[1], Thomas Kühn[2], Taina Yli-Juuti[1], Anton Laakso[3], Eemeli Holopainen[3,4], Douglas R. Worsnop[1,5], Annele Virtanen[1], and Harri Kokkola[1,3]

[1]Department of Technical Physics, University of Eastern Finland, Kuopio, Finland
[2]Weather and Climate Change Research, Finnish Meteorological Institute, Helsinki, Finland
[3]Atmospheric Research Centre of Eastern Finland, Finnish Meteorological Institute, Kuopio, Finland
[4]Center for the Study of Air Quality & Climate Change (CSTACC), Institute of Chemical Engineering Sciences, Foundation for Research and Technology – Hellas (FORTH/ICE-HT), Patras, Greece
[5]Aerodyne Research, Inc., Billerica, Massachusetts, United States

**Correspondence:** Harri Kokkola (harri.kokkola@fmi.fi)

**Abstract.**

Secondary organic aerosol (SOA) constitutes an important component of atmospheric particulate matter, with substantial influence on air quality, human health and the global climate. Volatility basis set (VBS) framework has provided a valuable tool for better simulating the formation and evolution of SOA where SOA precursors are grouped by their volatility. This is done in order to avoid the computational cost of simulating possibly hundreds of atmospheric organic species involved in SOA formation. The accuracy of this framework relies upon the accuracy of the volatility distribution of the oxidation products of volatile organic compounds (VOCs) used to represent SOA formation. However, the volatility distribution of SOA forming vapours remains inadequately constrained within global climate models, leading to uncertainties in the predicted aerosol mass loads and climate impacts. This study presents the results from simulations using a process-scale particle growth model and a global climate model, illustrating how uncertainties in the volatility distribution of biogenic SOA precursor gases affects the simulated cloud condensation nuclei (CCN). We primarily focused on the volatility of oxidation products derived from monoterpenes as they represent the dominant class of VOCs emitted by boreal trees. Our findings reveal that the particle growth rate and their survival to CCN sizes, as simulated by the process scale model, are highly sensitive to uncertainties in the volatilities of condensing organic vapours. Interestingly, we note that this high sensitivity is less pronounce in global scale model simulations, as the CCN concentration and cloud droplet number concentration (CDNC) simulated in the global model remain insensitive to a one order of magnitude shift in the volatility distribution of organics. However, a notable difference arises in the SOA mass concentration as a result of volatility shifts in the global model. Specifically, a one order of magnitude decrease in volatility corresponds to an approximate 13% increase in SOA mass concentration, while, a one order of magnitude increase results in a 9% decrease in SOA mass concentration over the boreal region. SOA mass and CCN concentrations are found to be more sensitive to the uncertainties associated with the volatility of semi-volatile compounds, with saturation concentrations of $10^{-1}$ µg/m$^3$ or higher, than the low-volatile compounds. This finding underscores the importance of having higher resolution in the semi-volatile bins, especially in global models, to accurately capture SOA formation. Furthermore, the

study highlights the importance of a better representation of saturation concentration values for volatility bins when employing a reduced number of bins in a global scale model. A comparative analysis between a finely resolved 9-bin VBS setup and a simpler 3-bin VBS setup highlights the significance of these choices. The study also indicates that radiative forcing attributed to changes in SOA over the boreal forest region is notably more sensitive to the volatility distribution of semi-volatile compounds than low-volatile compounds. In the 3-bin VBS setup, a ten-fold decrease in volatility of the highest volatility bin results in a shortwave instantaneous radiative forcing (IRF$_{ari}$) of -0.2 $\pm$ 0.10 Wm$^{-2}$ and an effective radiative forcing (ERF) of +0.8 $\pm$ 2.24 Wm$^{-2}$, while a ten-fold increase in volatility leads to an IRF$_{ari}$ of +0.05 $\pm$ 0.04 Wm$^{-2}$ and ERF of +0.45 $\pm$ 2.3 Wm$^{-2}$ over the boreal forest region. These findings underscore the critical need for a more accurate representation of semi-volatile compounds within global scale models to effectively capture the aerosol loads and the subsequent climate effects.

## 1   Introduction

Organic aerosol (OA) plays a critical role in atmospheric chemistry, comprising a significant fraction of sub-micrometer atmospheric aerosols (Jimenez et al., 2009). These atmospheric aerosol make up 20-90% of the total submicron aerosol loading (Jimenez et al., 2009; Hallquist et al., 2009), and they are crucial for both human health and the climate. OA includes Primary Organic Aerosol (POA) and Secondary Organic Aerosol (SOA), both of which are composed of a mixture of organic chemical species. POA is directly emitted into the atmosphere from a variety of sources, such as vegetation, biomass burning, and fossil fuel combustion (Spracklen et al., 2011). In contrast, SOA is formed in the atmosphere by the gas-phase oxidation of organic compounds to form products that subsequently condense into the aerosol phase (Kroll and Seinfeld, 2008). This condensation process largely depends on the saturation vapor concentration ($C_{sat}$) of the oxidized products. The formation of SOA is driven by the oxidation of a variety of Volatile Organic Compounds (VOCs) or Semivolatile Organic Compounds (SVOCs) present in the atmosphere, through reactions with oxidants such as hydroxyl radicals (OH), ozone (O$_3$), and nitrate radicals (NO$_3$) (Ng et al., 2017). The oxidation of these organic compounds can then lead to the formation of an extensive range of low-volatile and semi-volatile products that can then condense into the particle phase (Hallquist et al., 2009). Terpenes (e.g., $\alpha$-pinene and $\beta$-pinene) and isoprene are dominant sources of biogenic VOCs globally, while alkanes and aromatics (e.g., toluene and xylene) are the major anthropogenic VOCs (Ziemann and Atkinson, 2012). In the boreal ecosystem, monoterpenes account for the majority of VOC emissions, which yield a significant amount of global SOA, comprising more than half of the total biogenic SOA (Yu et al., 2021).

SOA, especially from biogenic origin, plays a significant role in Earth's climate, primarily through their influence on aerosol-cloud interactions and aerosol-radiation interactions (Scott et al., 2014; Yli-Juuti et al., 2021; Petäjä et al., 2022). In particular, the effect of SOA on clouds is mainly determined by the particles that grow to CCN sizes (typically 30 nm to 100 nm), primarily through the process of condensation (Pierce and Adams, 2007; Pierce et al., 2012). In addition to their effect on cloud properties, SOA can also affect climate directly by affecting radiative transfer of solar radiation in the atmosphere. SOA significantly scatters solar radiation, which can lead to a cooling effect on the Earth's surface (Shrivastava et al., 2017).

Even though there have been substantial improvements in understanding properties and formation mechanisms of biogenic SOA, its representation in the current climate models is still poorly constrained (Hodzic et al., 2016; Shrivastava et al., 2017; Tsigaridis and Kanakidou, 2018; Liu et al., 2021). One potential source of these uncertainties is associated with the complex and highly variable composition of SOA and its precursors (Zhu et al., 2017), which is influenced by a variety of different VOCs and their multiple oxidation pathways (Donahue et al., 2012). Although certain climate models have made progress in

simulating the formation of SOA, there are still uncertainties, particularly in aspects related to the representation of organic vapors. Given the wide spectrum of SOA precursor species, their concentrations, and compositions, several of the current climate models (for example, ECHAM-SALSA (European Centre Hamburg Model - Sectional Aerosol module for Large Scale Applications) (Mielonen et al., 2018), CESM2 (Community Earth System Model 2) (Tilmes et al., 2019), GEOS-CHEM (Goddard Earth Observing System - Atmospheric Chemistry) (Fritz et al., 2022), GFDL AM (Geophysical Fluid Dynamics

Laboratory's Atmosphere Model) (Zheng et al., 2023)) use volatility basis set (VBS) approach to represent SOA and to simulate the formation of SOA in the atmosphere. The VBS framework provides a systematic and computationally efficient way to represent the multitude of organic compounds and their varying volatilities, which determine how they partition between the gas and particle phase in the atmosphere (Donahue et al., 2006). In the VBS approach, SOA is treated as a mixture of organic compounds of varying volatility that are distributed among a set of discrete volatility bins based on their vapor pressure

(Donahue et al., 2011). The VBS framework aims to simplify the complex and often poorly understood processes involved in SOA formation. This simplification is achieved by combining compounds to groups based on their volatility reducing the complexity of chemical processes involved in the formation and aging of SOA (Donahue et al., 2006). The VBS approach has been shown to be effective at simulating the concentration and composition of SOA in the atmosphere and can be used to study the impacts of SOA on air quality and the climate (Tsimpidi et al., 2010; Jathar et al., 2017; Jiang et al., 2019).

Monoterpenes dominate the VOC emissions in boreal forested areas, covering 29% of forested land areas, thereby making this region a significant source of biogenic SOA (Rinne et al., 2009; Kayes and Mallik, 2020). Hence, the significance of monoterpenes and their oxidation products to understand global natural background aerosol burdens is evident. Several studies have demonstrated the importance of oxidation products of monoterpenes, and their vapour pressures, in the new particle formation and growth in a process level (e.g. Ehn et al. (2014); Tröstl et al. (2016); Kirkby et al. (2016); Yli-Juuti et al.

(2017); Lehtipalo et al. (2018); Roldin et al. (2019); Mohr et al. (2019)). However, the sensitivity of global model outputs (CCN concentrations, cloud droplet number concentration and radiative forcing) to uncertainties in volatilities of compounds or resolution (i.e. number of bins) in VBS representing monoterpene oxidation products remains inadequately studied.

    In this study, we used a process-scale growth model and a global aerosol-climate model to investigate the sensitivity of volatility distribution to SOA formation and cloud properties. The process model focused on examining the sensitivity of

particle growth rate and their survival across CCN size ranges to uncertainties in the volatilities of organic vapours in a process scale. One motivation for this study was to examine how process model results, which only take into account microphysical processes, translate to global scale, which also includes several processes that can buffer the changes SOA makes to the aerosol population. To understand how these sensitivities in the process scale manifest at a global scale, we utilized the global aerosol-climate model ECHAM-HAMMOZ coupled with aerosol microphysical model SALSA, which is the Sectional Aerosol module

for Large Scale Applications (Kokkola et al., 2018; Holopainen et al., 2020, 2022). This allowed us to study how sensitive the simulated SOA formation is to the assumptions of the volatility distributions of biogenic SOA precursors, specifically monoterpenes, over the boreal region. We also investigated how this sensitivity translates to sensitivities in simulated cloud properties as well as the radiative properties of the atmosphere. To study the sensitivity of SOA mass and CCN concentrations to uncertainties in the volatility distribution, we shifted the volatility of monoterpene oxidation products across all VBS bins by one order of magnitude. Additionally, we shifted the volatilities of individual VBS bins by one order of magnitude to assess

the effect of uncertainties in the volatility of each VBS bin. To investigate the sensitivity of model outputs to VBS resolution, we tested two setups using the VBS approach. The first setup describes the volatility of SOA precursors by grouping them into 9 volatility classes (hereafter referred to as 9-bin VBS setup) while the second one groups them into 3 volatility classes (hereafter referred to as 3-bin VBS setup). From these simulations, we analyzed how the assumptions and changes in volatility distribution affects SOA mass, CCN and cloud properties. Overall, this study aims to provide insights into the importance of

accurately representing the volatility distribution of organic aerosols when modelling the formation of SOA in a global climate model.

## 2    Methods

### 2.1    Particle growth model

We simulated the growth of nucleation mode particles and their survival to CCN size on process-scale with a Model for coagulation losses in nanoparticle growth (MCOLNAG). MCOLNAG specifically focuses on understanding the effect of uncertainties in volatility distribution to particle growth and CCN concentration. The model simulates growth of a monodisperse nucleation mode particle population due to condensation of vapors and the decrease of the nucleation mode particle number concentration due to coagulation. Condensation growth for other compounds than water is calculated based on transition regime mass flux

equation including the effect of particle motion and vapor molecule volume influencing at small sizes (Fuchs and Sutugin, 1970; Lehtinen and Kulmala, 2003). Water uptake by growing particles is calculated by assuming constant instantaneous equilibration between gas and particle phase. Particle phase is assumed to form an ideal solution, behave liquid-like, and no particle phase chemical reactions are included. Decrease in nucleation mode particle concentration due to coagulation scavenging to Aitken and accumulation modes and self-coagulation within nucleation mode are included. Aitken and accumulation mode par-

ticle diameters and number concentrations are set constant. Self-coagulation decreases nucleation mode number concentration but its impact to particle size is ignored, i.e. the nucleation mode particles grow in the model by condensation only.

In this study, condensing vapors included the organic vapors presented with a 9-bin VBS and water. Concentrations of each bin were defined by setting the total concentration of organic vapors and multiplying that with the stoichiometric coefficient of the respective bin (see Sect. 2.3 for stoichiometric coefficients). Particle number concentrations and diameters of Aitken and

accumulation mode as well as the initial number concentration of nucleation mode were set to spring-time median values at boreal forest measurement station Hyytiälä reported by Leinonen et al. (2022). Number concentrations were $1161 \, \mathrm{cm}^{-3}$ and $321 \, \mathrm{cm}^{-3}$ and diameters were $53 \, \mathrm{nm}$ and $170 \, \mathrm{nm}$ for Aitken and accumulation modes, respectively. Initial number concentra-

tion of nucleation mode was $539\,\mathrm{cm}^{-3}$ and initial diameter of the nucleation mode particles was $3\,\mathrm{nm}$. The molar mass of the organic compounds was $200\,\mathrm{g\,mol}^{-1}$, gas phase diffusion coefficient at $273.15\,\mathrm{K}$ was $5 \times 10^{-6}\,\mathrm{m}^2\,\mathrm{s}^{-1}$ from which the value at set temperature was calculated, mass accommodation coefficient was 1, particle density was $1200\,\mathrm{kg\,m}^{-3}$, and surface tension of the particle was $30\,\mathrm{mN\,m}^{-1}$. The simulations were performed at $298\,\mathrm{K}$.

## 2.2 Global aerosol-climate model ECHAM-SALSA

We used the global aerosol-climate model ECHAM-HAMMOZ (ECHAM6.3-HAM2.3)(Schultz et al., 2018) to study the sensitivity of SOA formation and cloud properties to changes in the volatility distribution of organics. ECHAM-HAMMOZ consists of the atmospheric general circulation model ECHAM6, which solves the equations of motion and continuity for the atmosphere using the spectral method. In this study, all simulations utilized the T63 spectral truncation in horizontal resolution, which corresponds to a grid spacing of roughly $1.9° \times 1.9°$ and 47 hybrid sigma-pressure levels for vertical resolution.

HAM (Hamburg Aerosol Module) (Kokkola et al., 2018; Tegen et al., 2019) in ECHAM-HAM simulates the life cycle of aerosols in the atmosphere, including their formation, growth, and removal processes. HAM includes a comprehensive treatment of aerosol microphysics, including the formation of SOA from gas-phase precursors, the growth of particles through coagulation, emissions of gases and aerosol, the removal of particles by dry and wet deposition, aerosol-radiation interactions, and aerosol-cloud interactions.

HAM offers two different options for modelling aerosol microphysics. One of the options is the modal aerosol module, called M7, which is designed to simulate the number and mass concentrations of different aerosol modes, based on their size and composition (Vignati et al., 2004; Stier et al., 2005). The other option is called the Sectional Aerosol module for Large Scale Applications (SALSA) (Kokkola et al., 2018), which uses a sectional approach to model aerosol microphysics. In this study, ECHAM-HAMMOZ is coupled with SALSA (Kokkola et al., 2018), employing a detailed sectional representation of aerosol microphysics. SALSA uses several discrete size classes to represent aerosol size distribution which is described more in Sect. 2.2.1. Out of the two options, only SALSA includes the VBS approach for describing SOA formation.

### 2.2.1 SALSA

SALSA is an advanced aerosol microphysical model that simulates the size distribution and chemical composition of atmospheric aerosols. SALSA uses a sectional approach where the aerosol size distribution is divided into 10 size classes in size space, ranging from $3\,\mathrm{nm}$ to $10\,\mathrm{\mu m}$. For particles larger than $50\,\mathrm{nm}$, the model includes parallel externally mixed size classes (Kokkola et al., 2018). SALSA includes a comprehensive treatment of aerosol microphysics, including nucleation, condensation/evaporation, coagulation, and hydration. SALSA simulated aerosol is also coupled to aerosol-cloud interactions as well as radiation, allowing for investigations of the impacts of aerosols on the Earth's radiative budget and thus the climate. Cloud droplet activation is solved using the parameterization by Abdul-Razzak and Ghan (2002) which calculates the fraction of activated particles in each size class. SALSA treats the chemical species: sulfate, organic carbon, black carbon, sea salt, and mineral dust. A detailed description of the model is given by Kokkola et al. (2018). However, recently SALSA has undergone further advancements to enhance its wet scavenging scheme introduced by Holopainen et al. (2020). SALSA has been used in

numerous studies at different spatial scales (e.g., Bergman et al. (2011); Andersson et al. (2015); Tonttila et al. (2017); Kühn et al. (2020); Miinalainen et al. (2021); Holopainen et al. (2022)) to investigate the behavior of atmospheric aerosols and their impacts on the climate.

### 2.2.2 SOA formation routine of SALSA

SALSA includes a comprehensive SOA parameterization based on the VBS framework (Stadtler et al., 2018; Mielonen et al., 2018). In this study, we have set up the VBS approach so that it categorizes VBS compounds into either nine or three different volatility bins. SOA is composed of both anthropogenic and biogenic VOC sources. SOA formation is represented by the partitioning of the oxidized organic compounds between the gas and particle phases based on their volatility. To estimate the partitioning of VOC oxidation products between the gas and particle phases, the Analytical Predictor of Condensation (APC) method, developed by Jacobson (2005) is used. The APC method calculates the partitioning of VBS species by solving the condensation equation

$$\frac{dC_{\mathrm{org},i}}{dt} = k_{m,i}(C_{\mathrm{org},i,\mathrm{surf}} - C_{\mathrm{org},\mathrm{gas}}), \tag{1}$$

numerically. This enables the estimation of non-equilibrium partitioning of each organic compound "org" between gas phase and each aerosol size class $i$ and thus their contribution to SOA formation. To avoid any oscillatory behaviours in condensation, we solve the condensation equations using five time steps to solve the condensation over one atmospheric model time step, with condensation solver time step length increasing logarithmically. Eq. (1) describes the rate of change of the gas-phase concentration of the organic compound, represented by $C_{\mathrm{org},i}$, with time ($dt$) and is a function of the mass transfer coefficient $k_{m,i}$, the surface equilibrium concentration of the organic compound in the particle phase $C_{\mathrm{org},i,\mathrm{surf}}$, and the gas-phase concentration of the organic compound. The calculation of the saturation concentration at the surface of a droplet is determined using

$$C_{\mathrm{org},i,\mathrm{surf}} = S'C_{\mathrm{org},i,\mathrm{surf}} = S'_i x_{\mathrm{org},i} C_{org,sat} \tag{2}$$

In Eq. (2), $S'_i$ represents the Kelvin effect, $x_{\mathrm{org},i}$ represents the mole fraction of the organic compound, and $C_{org,sat}$ represents the saturation concentration of the condensing compound. In these equations, concentrations are expressed as mole concentrations (mol/m$^3$), which are derived from mass concentration based values. The method assumes that the behavior of the condensing organic compounds in the condensed phase is ideal (Kokkola et al., 2014).

### 2.3 Formulation of the volatility distribution

ECHAM-SALSA normally uses a 3-bin VBS setup for monoterpene oxidation products with $C_{sat}$ values 0, 1 and 10 µg/m$^3$ and different stoichiometric parameters for low and high NOx conditions based on Pathak et al. (2007). In order to analyze the sensitivity of SOA formation on volatility distribution, we needed to implement a VBS parameterization for monoterpene oxidation products with more volatility bins than in the default model setup. Hence, we constructed the volatility distribution of SOA precursors based on previous studies conducted by Tröstl et al. (2016) and Hunter et al. (2017). Tröstl et al. (2016)

conducted a chamber experiment to determine the gas phase volatility distribution of $\alpha$-pinene oxidation products (Tröstl et al. (2016) Extended Data Figure 5b). Extremely low-volatility organic compound (ELVOC) concentrations were measured using a nitrate chemical ionization mass spectrometer (CIMS), while LVOC and SVOC concentrations were estimated based on the growth rates of nanoparticles. Hunter et al. (2017) measured organic concentrations in the particle and gas phases during summertime in a Ponderosa pine forest using five mass spectrometers. Here, we used their reported campaign-average volatility distribution (Hunter et al., 2017), but included only the gas phase data and excluded the fraction referring to particle phase. However, the results from Tröstl et al. (2016) for the upper end of LVOC range and for SVOCs are uncertain due to lower sensitivity of particle growth to these compounds, and they omitted saturation concentrations higher than $100\ \mu g/m^3$. In contrast, Hunter et al. (2017) did not detect gas phase ELVOCs, even though their existence has been established in $\alpha$-pinene laboratory systems and boreal forest. This suggests that their array of mass spectrometers had some deficiencies in detecting organic compounds at the lower end of volatilities. Therefore, the volatility distribution for this study was constructed by combining the volatility distributions from both Hunter et al. (2017) and Tröstl et al. (2016). Hunter et al. (2017) report volatilities at $298\ \mathrm{K}$ as the reference temperature for $C_{sat}$ while Tröstl et al. (2016) experiments were performed at $278\ \mathrm{K}$. In order to combine the two volatility distributions, the volatility distribution from Tröstl et al. (2016) was converted to $298\ \mathrm{K}$ assuming a vaporization enthalpy ($\Delta H_{\mathrm{vap}}$) of $30\ \mathrm{kJ\,mol}^{-1}$ (Farina et al., 2010). First, a logarithmic uniform distribution of compounds within each VBS bin was assumed (i.e. between $10^{n-0.5}$ - $10^{n+0.5}\ \mu g/m^3$ for the bin $C_{sat} = 10^n\ \mu g/m^3$) and the concentration of each bin was divided into 100 logarithmically uniformly distributed sub-bins. Third, the sub-bins were redistributed into traditional VBS bins at 298 K with $C_{sat}$ values of full orders of magnitude using the bin limits of $10^{n-0.5}$ - $10^{n+0.5}\ \mu g/m^3$ or the bin with $C_{sat} = 10^n\ \mu g/m^3$.

The two volatility distributions were combined by assuming that both distributions included all the compounds that belong to the bin $C_{sat} = 10^{-1}\ \mu g/m^3$ at 298 K. First, the concentration in each bin in both volatility distributions were normalized to the concentration in bin $C_{sat} = 10^{-1}\ \mu g/m^3$. Then the combined volatility distribution was constructed by selecting the volatility bins $C_{sat} \leq 10^{-1}\ \mu g/m^3$ from Tröstl et al. (2016) and the bins $10^{-1}\ \mu g/m^3 < C_{sat} < 10^1\ \mu g/m^3$ from Hunter et al. (2017). Finally, the total concentration in the constructed volatility distribution was normalized to 1, which provided a normalized distribution to use as a starting point to generate volatility basis sets with different numbers of bins.

In order to reduce the computational costs, the number of volatility bins was reduced by combining the bins $C_{sat} \leq 10^{-5}$ $\mu g/m^3$ into one non-volatile bin ($C_{sat} = 0$). This is a reasonable simplification since the growth process is insensitive to the saturation concentration of such extremely low-volatility compounds (Kokkola et al., 2014). This resulted in a 9-bin VBS with one non-volatile bin and eight bins where $10^{-4}\ \mu g/m^3 \leq C_{sat} \leq 10^3\ \mu g/m^3$. The stoichiometric coefficients for each volatility bin were calculated by multiplying the sum of the stoichiometric coefficients by the normalized concentration in the respective bin. The $C_{sat}$ values and the corresponding stoichiometric coefficients for the 9-bin VBS setup are given in Table 1.

The described 9-bin VBS setup is computationally very expensive to be used in a global sectional aerosol model framework such as SALSA. This is why the VBS setup of several global models, for instance, WRF-CHEM (Reyes-Villegas et al., 2022), CESM2 (Tilmes et al., 2019), and the previous version of ECHAM-SALSA (Mielonen et al., 2018) have simpler VBS representations. However, the implications of such a simplification have not previously been studied in a global model

**Table 1.** Overview of Saturation Concentrations and Stoichiometric Coefficients used for 9-Bin and 3-Bin VBS setups

| 9-Bin VBS | | 3-Bin VBS | | |
|---|---|---|---|---|
| $C_{sat}$ | $\alpha$ | $C_{sat}$ | $\alpha$ | **Bin** |
| 0 | 0.0038 | | | |
| $10^{-4}$ | 0.0029 | $4.8 \times 10^{-4}$ | 0.0126 | Bin 1 |
| $10^{-3}$ | 0.0059 | | | |
| $10^{-2}$ | 0.0146 | | | |
| $10^{-1}$ | 0.016 | $5.48 \times 10^{-1}$ | 0.0639 | Bin 2 |
| $10^{0}$ | 0.0333 | | | |
| $10^{1}$ | 0.1028 | | | |
| $10^{2}$ | 0.1456 | $5.32 \times 10^{2}$ | 0.497 | Bin 3 |
| $10^{3}$ | 0.2491 | | | |

framework. To compare the impact of the number of volatility bins on the SOA formation and the climate, a 3-bin VBS setup was developed based on the already formulated 9-bin VBS setup. The 3-bin VBS setup lumped together the adjacent bins from the 9-bin VBS setup to form three bins. Bin1 consisted of the three adjacent bins with $C_{sat}$ values of 0, $10^{-4}$, and $10^{-3}$

$\mu g/m^3$; Bin2 included $C_{sat}$ values of $10^{-2}$, $10^{-1}$, and $10^{0}$ $\mu g/m^3$; and Bin3 comprised $C_{sat}$ values of $10^{1}$, $10^{2}$, and $10^{3}$ $\mu g/m^3$. The stoichiometric coefficients for the 3-bin VBS setup were obtained by summing the stoichiometric coefficients of the three bins considered from the 9-bin VBS setup. Consequently, the total production of condensable organics resulting from the oxidation of monoterpene remained unchanged between the 9-bin VBS setup and the 3-bin VBS setup. In the 3-bin VBS setup, we calculated the saturation concentration for each bin as the concentration-weighted arithmetic average of the three

lumped bins from the 9-bin VBS setup. The $C_{sat}$ values and the corresponding stoichiometric coefficients for the 3-bin VBS setup are given in Table 1.

## 2.4  ECHAM-SALSA simulations

A series of simulations was conducted using the global aerosol-climate model ECHAM-SALSA to investigate the effect of shift in volatility of monoterpene oxidation products on aerosol properties and radiative forcing. Two different VBS setups were

used, namely 9-bin VBS setup and 3-bin VBS setup, to evaluate the sensitivity of the model to the number of volatility bins. Three different simulations were performed using the 9-bin VBS setup to assess the impact of uncertainties in the volatility distribution by shifting the volatility of monoterpene oxidation products by one order of magnitude. The simulations were conducted for the original volatility (VBS×1), increased volatility (VBS×10), and decreased volatility (VBS×0.1). Another set of six different simulations were carried out using the 3-bin VBS setup to study the effect of uncertainties associated

with the volatilities of individual VBS bins where the volatility of each VBS bin was shifted by one order of magnitude. A summary of all the simulations are given in Table 2. We studied the sensitivity of SOA mass, CCN, and radiative forcing to the

volatility assumptions across all simulations. Number concentration of particles greater than $100\,\mathrm{nm}$ (N100) in diameter was used as a proxy for the CCN concentration (Clarke and Kapustin, 2010). This study focused solely on the shift in oxidation products of monoterpenes, as they are the most abundant VOCs emitted by the boreal trees (Rinne et al., 2009). Furthermore, the highest emissions of monoterpene from the boreal trees generally occur during the summer season (Vanhatalo et al., 2020), and therefore, we specifically studied the summer season (June, July and August) of the simulation year 2010. Consequently, all the findings and results from this study are specific to the boreal forest region in the Northern Hemisphere, where the summer season is characterized by intense monoterpene emissions and their subsequent impact on aerosol dynamics.

**Table 2.** Summary of Experiments

| Experiment | VBS Setup | Volatility Shift | Experiment Description |
|---|---|---|---|
| 1 | 9-bin | VBS×1 | Original volatility |
| 2 | 9-bin | VBS×10 | Increased volatility by one order of magnitude |
| 3 | 9-bin | VBS×0.1 | Decreased volatility by one order of magnitude |
| 4 | 3-bin | bin1×10 | Increased volatility of bin1 by one order of magnitude |
| 5 | 3-bin | bin1×0.1 | Decreased volatility of bin1 by one order of magnitude |
| 6 | 3-bin | bin2×10 | Increased volatility of bin2 by one order of magnitude |
| 7 | 3-bin | bin2×0.1 | Decreased volatility of bin2 by one order of magnitude |
| 8 | 3-bin | bin3×10 | Increased volatility of bin3 by one order of magnitude |
| 9 | 3-bin | bin3×0.1 | Decreased volatility of bin3 by one order of magnitude |

All the simulations employed emission data from Community Emissions Data System (CEDS) for the anthropogenic emissions (Hoesly et al., 2018). In addition, we used the biomass burning emissions from Biomass Burning for Model Intercomparison Projects (BB4MIPs) inventory (Van Marle et al., 2017). The simulations were designed to allow the model atmospheric circulation to evolve freely while using fixed sea surface temperature (SST) and sea ice cover (SIC). Monthly mean climatologies from the Atmospheric Model Intercomparison Project (AMIP) provided the SST and SIC values (Taylor et al., 2012). To evaluate the impact of different assumed volatility distributions on the simulated Earth's radiation balance, we calculated the shortwave Effective Radiative Forcing (ERF) suggested by Forster et al. (2016), which is the difference in net top-of-atmosphere (TOA) radiative fluxes between simulations with different volatility distributions and the original volatility distribution. Furthermore, we calculated the shortwave Instantaneous Radiative Forcing due to aerosol-radiation interactions ($IRF_{ari}$) in ECHAM-SALSA by performing a double call to the radiation scheme with and without the aerosol perturbation as described in Collins et al. (2006). $IRF_{ari}$ differentiates the direct radiative effect of aerosols from the impact of aerosols on circulation and cloudiness which corresponds to the difference in net TOA radiative flux solely due to the absorption and scattering of aerosols without any contributions from adjustments. Additionally, we calculated one-sigma standard deviation across different grid cells over the boreal forests, indicating the variability within the boreal forest regions.

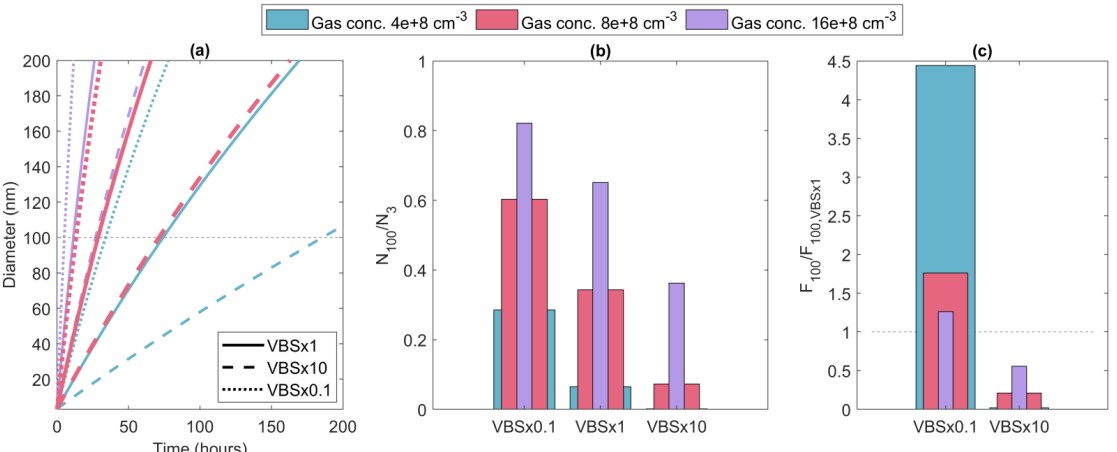

**Figure 1. (a)** Particle diameter as a function of time in MCOLNAG simulation with base case (solid lines) and volatilities shifted by one order of magnitude (dashed and dotted lines) in simulations with three different total organic vapor concentrations (indicated by line color), **(b)** fraction of nucleation mode particles reaching from initial size $3\,\mathrm{nm}$ to $100\,\mathrm{nm}$ in diameter, and **(c)** fraction of nucleation mode particles surviving to $100\,\mathrm{nm}$ (F100 = N100/N3) relative to the base case simulation.

## 3 Results

### 3.1 Particle growth model MCOLNAG

Figure 1 shows the particle growth simulated with MCOLNAG with the base case VBS and with the volatilities shifted higher or lower values by an order of magnitude as well as the fraction of nucleation mode particles that survives from the initial $3\,\mathrm{nm}$ population up to $100\,\mathrm{nm}$ in diameter. To approximate the conditions at boreal forest site Hyytiälä, the total organic concentration for the simulations was set to $8 \times 10^8\,\mathrm{cm}^{-3}$ (red lines and bars in Fig. 1). This led to particle growth rate of $3.7\,\mathrm{nm\,h}^{-1}$ for the diameter range of $3\text{-}20\,\mathrm{nm}$ which is in line with the median spring-time growth rates reported from

Hyytiälä (Yli-Juuti et al., 2011). Fraction of nucleation mode particles that survived to $100\,\mathrm{nm}$ increased by 76% and decreased by 79% when volatilities were shifted to lower and higher values, respectively. When volatilities were shifted to lower values, each bin (except for the non-volatile bin) became more prone to condense and more material became enough low-volatile to contribute substantially to particle growth and, therefore, particles grew faster, leading to less coagulation loss due to shorter growth time to $100\,\mathrm{nm}$. Shifting volatilities to higher values, reversely, led to each bin (except for the non-volatile bin) being

less prone to condense and less material existing that can condense to a significant degree to particle phase, slower particle growth and more coagulation losses. In such simplified process level model, the effect from shifting the volatility arises directly from the competition between particle growth and coagulation loss, and therefore the base case organic concentrations affect the sensitivity to shifting of volatility. To demonstrate the influence of total vapor concentration on the effect of shifting the volatilities, two additional sets of simulations are presented in the Fig. 1 with total vapor concentrations corresponding

half or double of the initially set value. With total organic vapor concentration decreased by 50%, the fraction of nucleation

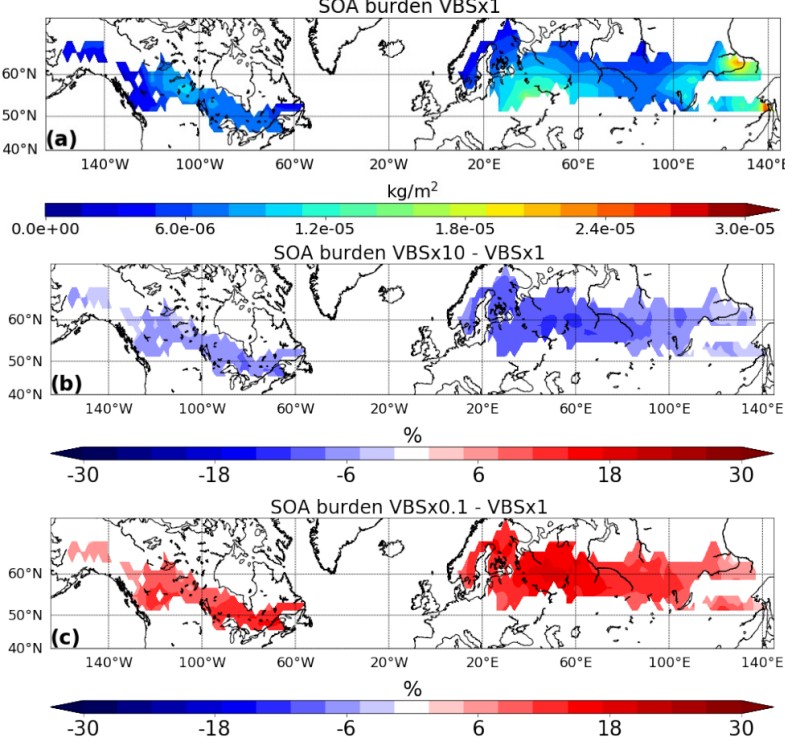

**Figure 2.** Simulated SOA burden of **(a)** VBS×1 and the relative difference between **(b)** VBS×10 and **(c)** VBS×0.1 with respect to VBS×1, focusing specifically on boreal forest regions to emphasize the sensitivity to monoterpene SOA.

mode particles reaching 100 nm was overall smaller and got relatively more sensitive to the shifting of volatility. Vice versa, with higher vapor concentration larger fraction of the nucleation mode particles survived to 100 nm size and their survival probability was relatively less sensitive to the volatility shift. Overall, these results demonstrate that at the process level, the particle growth rate and their survival to CCN sizes is sensitive to the uncertainties in volatilities of the condensing organic vapors.

## 3.2 Global aerosol-climate model ECHAM-SALSA

### 3.2.1 Sensitivity analysis using 9-bin VBS setup

First, we investigated how sensitive SOA burden is to the shifting of volatility of the monoterpene oxidation products when using the 9-bin VBS setup. In Fig. 2, we present the simulated SOA burden from the base case volatility (VBS×1) and the relative difference in the mean SOA burden between the simulations with shifted volatilities (VBS×10 and VBS×0.1) and base case volatility (VBS×1) for the summer period of the simulation year 2010. As expected, increase in volatility led to a decrease in the SOA burden, while a decrease in volatility resulted in an increase in the SOA burden. A ten-fold increase in volatility

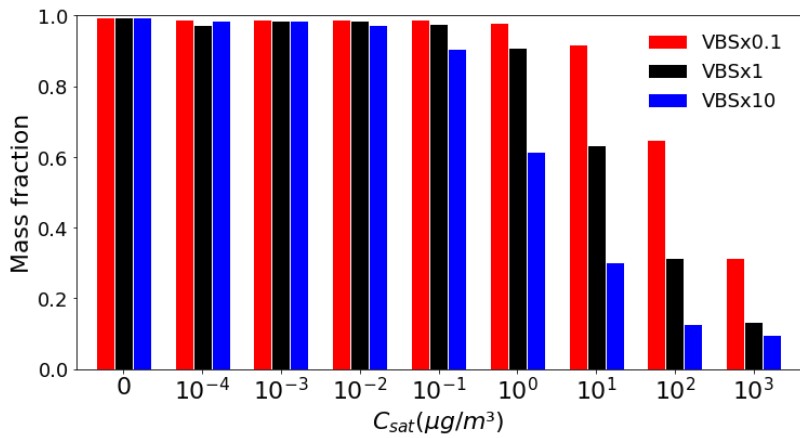

**Figure 3.** Fraction of particle phase mass concentration close to the ground level (particle/(particle+gas)) in each VBS bins with changes in volatility, depicted for the boreal forest region. The x-axis values show the $C_{\text{sat}}$ of that bin in the VBS×1 simulation

led to an average reduction of 9% in the SOA mass burden, while a ten-fold decrease in volatility resulted in a 13% increase in the SOA mass burden over the boreal region. These changes in SOA burden are particularly noticeable over Scandinavia and throughout Central Russia.

The changes in SOA mass burden with changes in volatility are consistent with the corresponding changes in the behaviour of gas-particle partitioning of the organic compounds. As the volatility of SOA decreases, the organic compounds become more likely to condense onto particles, causing more organic compounds to exist in the particle phase, while when the volatility of SOA increases,the organic compounds are more likely to exist in the gas phase. To analyze how sensitive the SOA formation is to the shifting of volatility, we calculated the fraction of mass partitioned in the particle phase for the three volatility assumptions (VBS×0.1, VBS×1, VBS×10) for each volatility bin (see Figure 3). It is notable that almost all of the total mass in the lower volatility bins with $C_{\text{sat}}$ values ranging from 0 to $10^{-1}$ μg/m$^3$ is in the particle phase. This suggests that the SOA mass formed from these volatility bins is not sensitive to a reasonable uncertainties in their volatilities. However, the biggest differences in gas-aerosol partitioning is seen in bins with $C_{\text{sat}}$ values ranging from $10^0$ to $10^3$ μg/m$^3$, i.e. bins which are not fully partitioned to either gas or particle phase. This means that the uncertainties associated with the volatility of these semi-volatile bins make a notable contribution to the sensitivity of gas-particle partitioning to the shift in volatility distribution.

In order to investigate how the uncertainty in volatility distribution affects the simulated CCN and cloud properties, we analyzed how N100 and CDNC are affected when volatilities are shifted by one order of magnitude. Fig. 4 shows the simulated burden of N100 for the VBS×1 simulation and the relative differences between simulations with shifted volatilities (VBS×10 and VBS×0.1) and VBS×1 simulation. Similar to the SOA burden, increasing the volatility resulted in a decrease in N100, while decreasing the volatility led to an increase in N100. The shift in volatility had a smaller impact on the CCN burden than the SOA mass burden across the study region. The relative differences ranged from -2% to -7% for the VBS×10 and 1% to 5% for the VBS×0.1 compared to the VBS×1 simulation. Similar to the changes in SOA mass burden, increasing

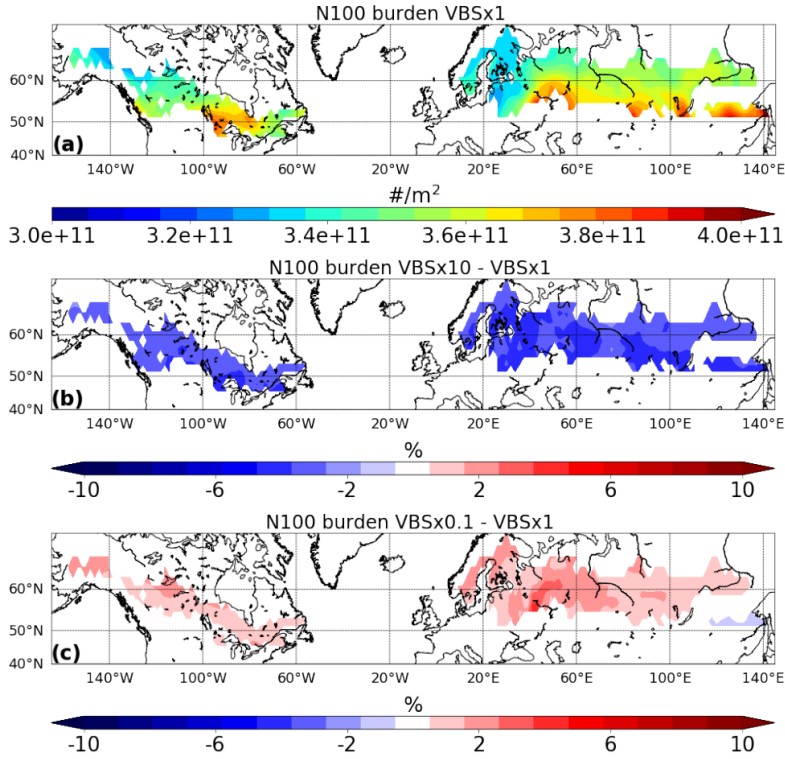

**Figure 4.** Simulated CCN burden of **(a)** VBS×1 and the relative difference between **(b)** VBS×10 and **(c)**VBS×0.1 with respect to VBS×1, focusing specifically on boreal forest regions to emphasize the sensitivity to monoterpene SOA.

volatilities reduces N100 and decreasing volatilities increases N100 almost over the whole region. However, specific pressure
levels exhibited a reduction in N100 with decrease in volatility at certain parts of the study region, particularly in regions that
are strongly affected by anthropogenic emissions (see Supplementary Material, Figure S1).

Figure 5 depicts the average vertical profile of SOA mass concentrations, N100 and CDNC for the original volatility simulation (VBS×1) and the simulations with shifted volatilities (VBS×10 and VBS×0.1). The mean relative difference in SOA mass concentration is found to be approximately -9% for VBS×10 and +13% for VBS×0.1 with respect to VBS×1 as seen
in Fig. 5a. Fig. 5b depicts that the mean relative difference in N100 is nearly -3% for VBS×10 and +2% for VBS×0.1 with respect to VBS×1. The changes are biggest at heights of typical boundary layer cloud base heights over the boreal regions thus having an impact on cloud activation.

As shown in Fig. 5c, the changes in CDNC with changes in volatility are consistent with the behaviour of N100 with a shift in volatility. There is a very small effect of the shift in volatility on the mean CDNC analyzed over the studied region. This suggests
that the change in the concentration of CCN particles due to a shift in volatility does not lead to a considerable change in the concentration of cloud droplets. This could be due to the fact that changes in CDNC are influenced by a combination of factors beyond CCN concentration alone, including cloud microphysics, meteorological conditions, and the aerosol composition.

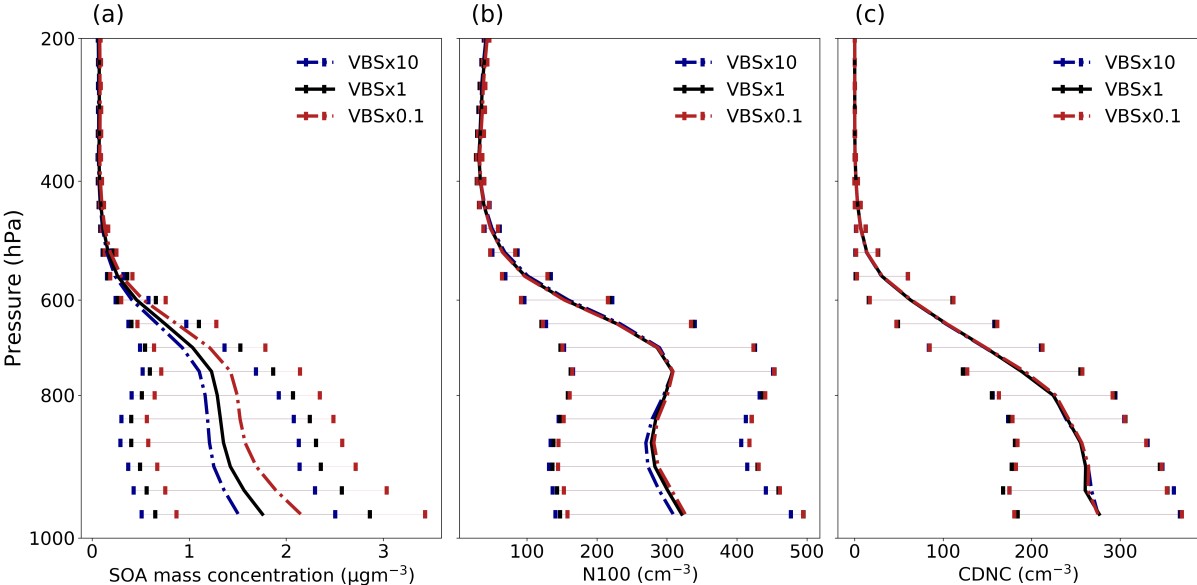

**Figure 5.** Mean vertical profile of **(a)** SOA mass concentrations , **(b)** N100, and **(c)** CDNC (Grids with cloud fraction $\leq$ 0.95 are excluded from the calculation of CDNC) at ambient conditions from volatility shifting simulations using 9-bin VBS setup over the boreal forest region. The error bar depicts one-sigma standard deviation of the data calculated across different grid cells over each model levels.

The observed shifts in SOA mass concentration and particle number concentration could be attributed to the partitioning behaviour of volatile compounds within the aerosol population. Specifically, low volatility compounds, which play a significant role in the growth of smallest particles to CCN sizes, tend to partition to the particle phase across all VBS setups as seen in Fig. 3. Notably, the biggest differences in partitioning occur for VBS bins with higher volatilities, which have more influence on particle mass in larger particles already in CCN sizes. Hence, while overall mass changes with volatility shifts, the CCN concentration and CDNC may remain relatively unchanged.

### 3.2.2 Comparison between 9-bin VBS setup and 3-bin VBS setup

Since increasing the number of volatility bins increases the computational cost of a global model, we also investigated how well a 3-bin VBS setup compares against a more accurately resolved 9-bin VBS setup. We assessed the sensitivity of SOA burden, N100 and CDNC comparing the otherwise identical simulations using 9-bin and 3-bin VBS setups. We applied the arithmetic mean to calculate volatility in the 3-bin VBS setup from the 9-bin VBS setup. Based on these simulations, the 9-bin VBS setup has a higher SOA burden, up to 20% more than the 3-bin VBS setup. It should be noted that the total production of condensable organics from monoterpene oxidation between both the VBS setups remains unchanged. At the ground level, the 9-bin VBS setup has 18% higher SOA mass concentration than the 3-bin VBS setup as shown in Fig. 6a. The difference in mass concentration between the two VBS setups primarily comes from the variations in mass concentration within the bin3 (third bin) of the 3-bin VBS setup compared to the cumulative mass concentration in the corresponding bins of the 9-bin VBS

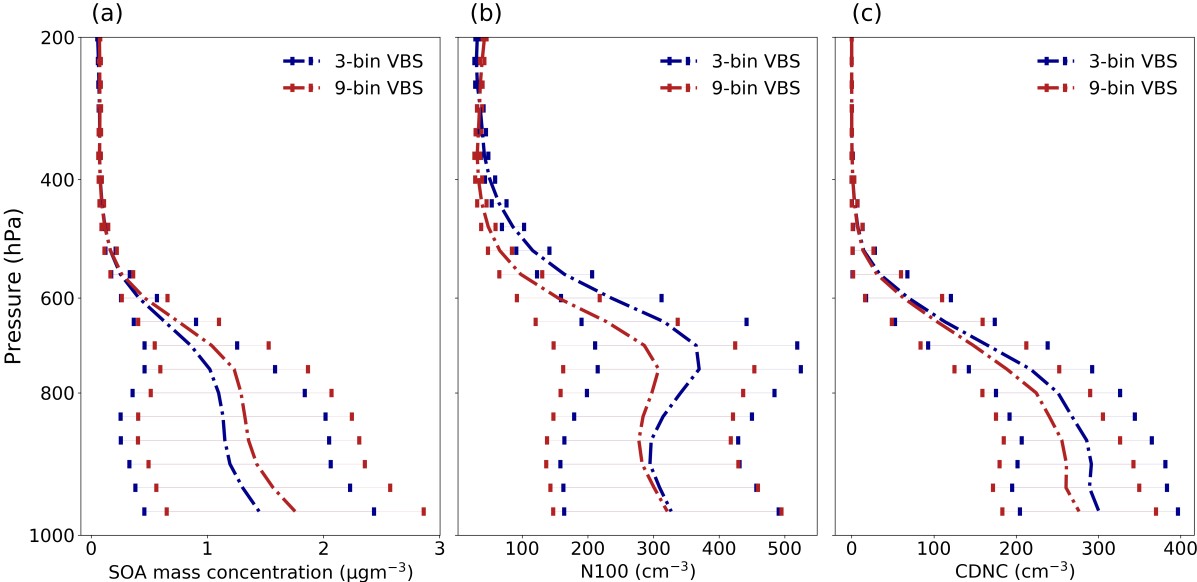

**Figure 6.** Mean Vertical profile of **(a)** SOA mass concentrations, **(b)** N100, and **(c)** CDNC (Grids with cloud fraction $\leq 0.95$ are excluded for the calculation of CDNC) at ambient conditions from the VBS$\times$1 simulation using 9-bin and 3-bin VBS setups over the boreal forest region. The error bar depicts one-sigma standard deviation of the data calculated across different grid cells over each model levels.

setup (See Supplementary Material Fig. S2). This results in a lower SOA mass concentration observed in the 3-bin VBS setup.
However, It's worth noting that the mass concentrations in the other two lower volatile bins in the 3-bin VBS setup remains consistent with the cumulative mass concentrations in their corresponding bins from 9-bin VBS setup. This differences between the two setups highlights the sensitivity of their configurations to the selection of volatilities assigned to each bin in the 3-bin VBS setup.

There is also a notable difference in the N100 concentration between the 9-bin VBS setup and 3-bin VBS setup as illustrated
in Fig. 6b. The difference in N100 between the 9-bin VBS setup and 3-bin VBS setup is in the opposite direction as compared to the SOA mass. i.e., 3-bin VBS setup has higher N100 concentration than the 9-bin VBS setup. Specifically, at around $500\,\mathrm{hPa}$, the 3-bin VBS setup has approximately 70% higher N100 compared to the 9-bin VBS setup. However, the difference in N100 between the two setups is only approximately 5% close to the ground level. In other words, VBS with lower number of volatility bins can result in higher N100 compared to the VBS with a higher number of volatility bins, although the SOA
mass shows an opposite behaviour. This could be because a lower number of volatility bins in VBS leads to a broader volatility range being assigned to each bin, which can lead to a higher concentration of SOA particles. Fig. 6c demonstrates that there is an increase in CDNC in the 3-bin VBS setup when compared with the 9-bin VBS setup. This is in line with the higher N100 observed in the 3-bin VBS setup. The higher N100 in the 3-bin VBS setup resulted in an approximate 10% increase in CDNC compared to the 9-bin VBS setup. It is worth noting that we also explored the geometric mean to calculate the volatility of

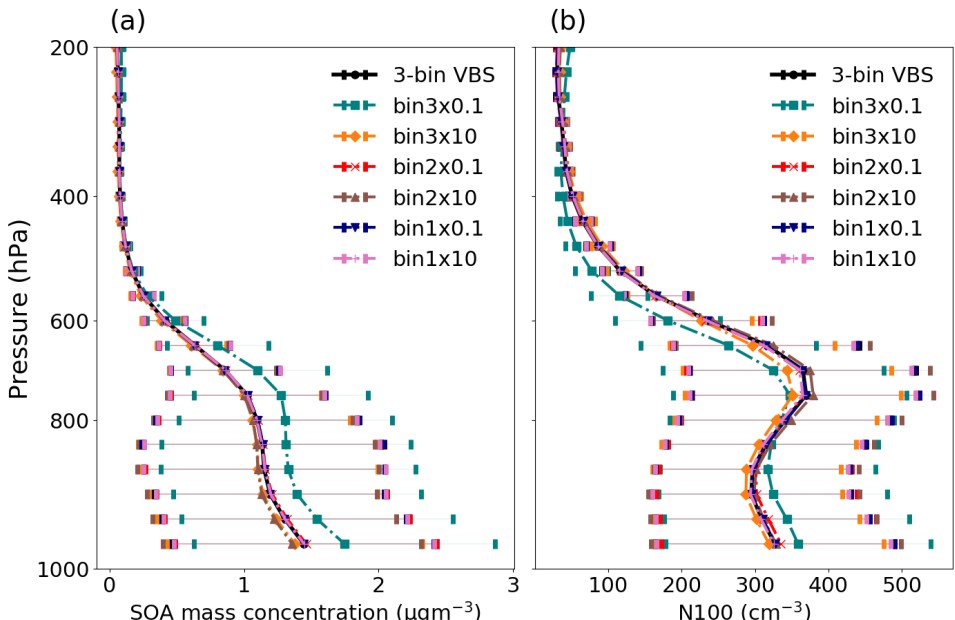

**Figure 7.** Mean vertical profile of **(a)** SOA mass concentrations and **(b)** N100 at ambient conditions when the VBS bins are shifted individually using the 3-bin VBS setup over the boreal forest region. The error bar depicts one-sigma standard deviation of the data calculated across different grid cells over each model levels.

3-bin VBS setup based on the 9-bin VBS setup, which resulted in better SOA mass matching between the two setups but worse matching for N100 (see Supplementary Material, Fig. S3).

### 3.2.3 Sensitivity of N100 to the volatility of individual VBS bins

To better understand the sensitivity of SOA mass and CCN concentration to uncertainties in the volatilities of individual VBS bins, a series of simulations were conducted in which the volatility of one VBS bin was shifted by one order of magnitude at a time, while the volatilities of other bins were kept unchanged. These simulations were performed in the global aerosol-climate model ECHAM-SALSA using the 3-bin VBS setup. Each bin in the 3-bin VBS setup is represented as bin1, bin2 and bin3 in the order of increasing volatility.

Figure 7 depicts the effect of a one order of magnitude shift in volatility of individual VBS bins on the SOA mass concentration and N100 over the studied region. The volatilities in the three bins: bin1, bin2 and bin3 are $4.8 \times 10^{-4}$, $5.48 \times 10^{-1}$ and $5.32 \times 10^{2}$ µg/m$^3$ respectively as described in Sect. 2.3. The change in bin1 (the lowest volatile class) doesn't seem to have any effect on either SOA mass concentration or N100. However, the increase in the volatility of bin2 led to approximately 7% decrease in SOA mass and 3% decrease in N100 and a decrease in volatility lead to nearly 4% increase in SOA mass and 3% increase in N100. Bin3 (the highest volatile class) is found to be the most sensitive VBS class to the changes in volatility. An increase of one order of magnitude in the volatility of bin3 led to a mean decrease of around 5% SOA mass and 2% N100 while

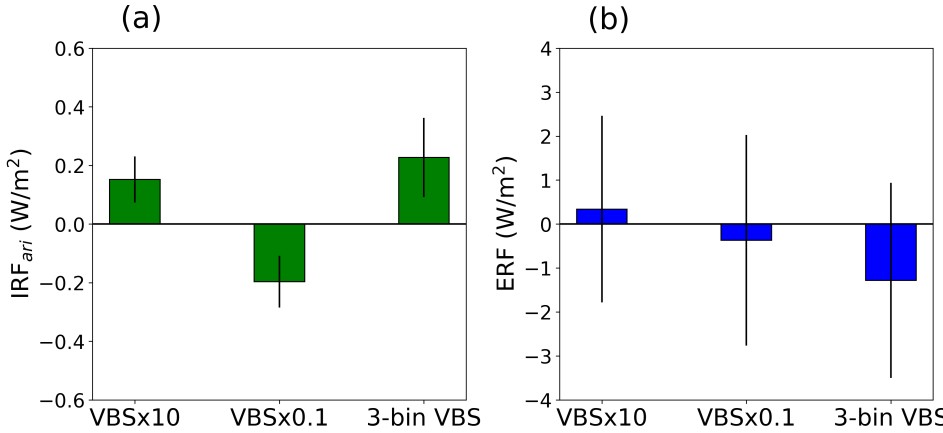

**Figure 8.** Summer Mean **(a)** TOA IRF$_{ari}$, **(b)** ERF for the shift in volatility by one-order of magnitude with respect to the base volatility using the 9-bin VBS setup, and the base volatility in 3-bin VBS setup relative to that in the 9-bin VBS setup, estimated over the boreal forest region. Error bar depicts one-sigma standard deviation of the data calculated across different grid cells.

a decrease in the volatility of bin3 led to approximately 20% increase in SOA mass and 9% increase in N100. When shifting the entire 3-bin VBS distribution, a ten-fold increase in volatility led to a notable 24% increase in SOA mass and a ten-fold decrease in volatility resulted in 8% reduction in SOA mass concentration. As the bin3 corresponds to the semi-volatile class in the VBS distribution, a significant amount of organic matter in this bin resides both in the gas and in the particle phase. Bin3 bundles the 9-bin setup bins with $C_{sat}$ values of $10^{-2}$, $10^{-1}$, and $10^0$ µg/m$^3$ (see Fig. 3). Out of these three bins, only the

highest volatility bin is sensitive to the shift in volatility where as combining them in one bin in the 3-bin VBS setup makes the $10^{-2}$–$10^0$ µg/m$^3$ volatility range sensitive to the assumed mean volatility of the bin (see Supplementary Material, Fig. S4). This is also the reason why shifting one bin in the 3-bin VBS setup affects the change in N100 and SOA mass shown in Fig. 7 than shifting the whole 9-bin distribution shown in Fig. 5. Overall, this test emphasizes the need of carefully selecting the volatilities of VBS bins when fewer bins are used in models so that it would reproduce a more highly resolved VBS bin setup.

**3.2.4    Effect of volatility distribution on Radiative forcing**

As discussed, SOA influences radiative forcing through both direct and indirect effects. According to O'Donnell et al. (2011), the estimated global mean SOA direct effect is -0.31 Wm$^{-2}$ while the indirect effect is +0.23 Wm$^{-2}$. However, it's important to note that a substantial uncertainty exists among different models, with a range of up to 1 Wm$^{-2}$ in the radiative effects of SOA, particularly for the first aerosol indirect effect (Zhu et al., 2017). Here, we assessed how the sensitivities in simulated aerosol

and cloud properties to the assumptions in volatility distribution affect the radiative properties of the simulated atmosphere. Specifically, we explored the radiative forcing due to aerosol-radiation and aerosol-cloud interactions. This section presents the quantified shortwave IRF$_{ari}$ and shortwave ERF for different volatility shifting simulations employing different number of volatility bins. Fig. 8 shows the IRF$_{ari}$ and ERF for one order of magnitude volatility shift with respect to the original volatility

using the 9-bin VBS setup and between the 3- and 9-bin VBS setups from the VBS×1 simulation. The decrease in SOA burden due to one order of magnitude increase in volatility contributes to a positive RF with respect to the original volatility in 9-bin VBS setup. Similarly, an increase in SOA burden due to a corresponding decrease in volatility leads to a more negative RF compared to that of the original volatility distribution. The $IRF_{ari}$ is found to be +0.16 ± 0.07 $Wm^{-2}$ and -0.2 ± 0.08 $Wm^{-2}$ for VBS×10 and VBS×0.1 simulations respectively with respect to the VBS×1 simulation, while ERF is +0.35 ± 2.1 $Wm^{-2}$ and -0.4 ± 2.3 $Wm^{-2}$ for VBS×10 and VBS×0.1 simulations respectively, relative to the VBS×1 simulation. Additionally, we estimated $IRF_{ari}$ and ERF from a scenario with the SOA shut off relative to the VBS×1 simulation. The $IRF_{ari}$ of SOA over the boreal forested region is -0.18 $Wm^{-2}$, and the ERF is -1.03 $Wm^{-2}$, highlighting the significant cooling effect of SOA in this region.

On the other hand, the base volatility in 3-bin VBS setup results in a positive $IRF_{ari}$ and negative ERF with respect to the base volatility in 9-bin VBS setup. It is because the 3-bin VBS setup has lower SOA mass and higher particle number concentration as compared to the 9-bin VBS setup. The summer mean $IRF_{ari}$ and ERF for the 3-bin VBS setup are found to be +0.25 ± 0.1 $Wm^{-2}$ and -1.25 ± 2.2 $Wm^{-2}$ respectively, relative to the base volatility in 9-bin VBS setup. The positive $IRF_{ari}$ and negative ERF in the 3-bin VBS setup are more pronounced over Russia while least over the northern US is shown in Supplementary Material, Fig. S5.

The results in Fig. 9 show the $IRF_{ari}$ and ERF calculated for simulations using a 3-bin VBS setup, where the volatility of individual volatility bins are shifted relative to the base volatility. The findings indicate that the changes in radiative forcings are consistent with the difference in SOA mass and N100 due to the shift in the volatility of individual VBS bins. The results reveal that the distribution of organic aerosol volatility substantially influences radiative forcing, encompassing both the $IRF_{ari}$ and ERF components of radiative flux. Similar to SOA mass and N100, the shift in volatility of the highest volatility bin (bin3) causes the largest effect in radiative forcing. The decrease in volatility of bin3 by one order of magnitude (bin3×0.1) leads to a cooling effect of -0.2 ± 0.1 $Wm^{-2}$ in $IRF_{ari}$ and a warming effect of 0.8 ± 2.24 $Wm^{-2}$ in ERF relative to the base volatility in the 3-bin VBS setup. This negative $IRF_{ari}$ and positive ERF from bin3×0.1 are attributed to the lower number concentrations and higher SOA mass as compared to the base volatility using the 3-bin VBS setup. Conversely, increasing the volatility of bin3 by one order of magnitude (bin3×10) induces a warming effect for both $IRF_{ari}$ and ERF. The corresponding values for the increased volatility scenario are 0.05 ± 0.04 $Wm^{-2}$ for $IRF_{ari}$ and 0.45 ± 2.3 $Wm^{-2}$ for ERF. While, the shift in the volatility of bin2 and bin1 have minor effects on the radiative forcing.

# 4 Conclusions

This study demonstrates the importance of accurate representation of semi-volatile organics in a global scale model to simulate aerosol-climate interactions. The volatility distribution of organics is crucial especially because it determines the partitioning of these compounds between the gas and particle phases. The partitioning process is essential for accurately simulating aerosol formation and growth processes in the atmosphere (Williams et al., 2010). In this study, we conducted a series of simulations using the process-scale model MCOLNAG and global aerosol-climate model ECHAM-SALSA to particularly examine how

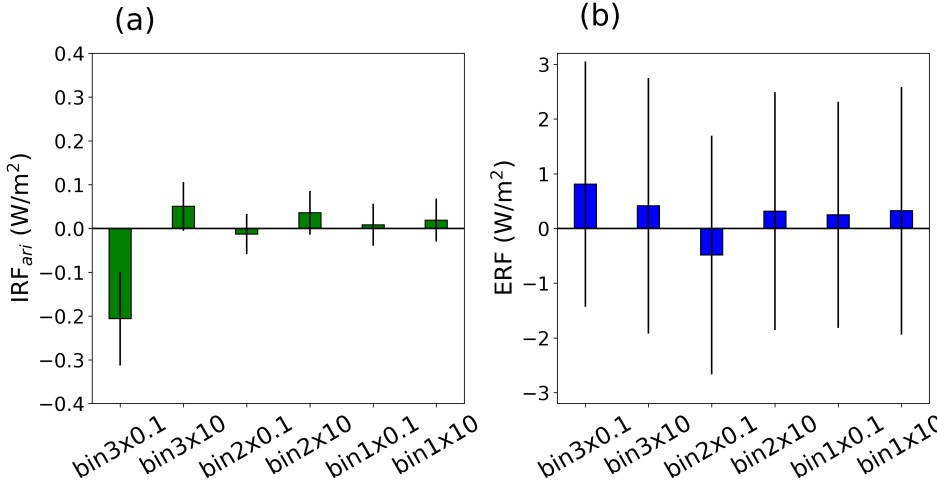

**Figure 9.** Summer Mean TOA **(a)** $IRF_{ari}$, **(b)** ERF for the shift in volatility of individual volatility bins with respect to the base volatility using the 3-bin VBS setup estimated over the boreal forest region. Error bar depicts one-sigma standard deviation of the data calculated across different grid cells.

sensitive the CCN is to the volatility assumptions of organic compounds. Although the process model simulations show a high sensitivity of CCN to the uncertainties associated with the volatilities of condensing organic vapours, the global model simulations with the highly-resolved 9-bin VBS setup show that N100 and CDNC are insensitive to a one order of magnitude
shift in volatility. However, a notable difference was observed in the global model simulated SOA mass burden. It was also found that nearly all of the total mass in the lower volatility bins of the 9-bin VBS setup, with $C_{sat}$ values up to $10^{-1}$, remained in the particle phase while, the bins with $C_{sat}$ values from $10^{0}$ contributed to the sensitivity of gas-particle partitioning. This behavior of gas-particle partitioning indicates that the lower volatility bins are insensitive to reasonable uncertainties in their volatilities, whereas the higher volatilities with $C_{sat}$ values of $10^{-1}$ μg/m$^3$ or higher are highly sensitive to their volatilities.
This suggests that the semi-volatile bins in the VBS setups require higher resolution than the low volatile compounds in a global model. Essentially, if a global model needs to represent organic compounds in fewer volatility bins, then ideally, the lower volatile bins with $C_{sat}$ values up to $10^{-1}$ can be lumped together into a single bin, while the remaining bins can have finer representation. Additionally, the global model analysis indicated that the SOA mass burden, N100 and radiative forcing were most sensitive to the uncertainties associated with the volatility of semi-volatile bins rather than the low-volatile bins. The
simulations also show that the $IRF_{ari}$ is sensitive to uncertainties in volatility in the 9-bin VBS setup while, the ERF becomes sensitive only if the simplified 3-bin VBS setup is used, as evident from the larger uncertainty associated with ERF changes. For example, in the VBS×10 scenario, the $IRF_{ari}$ is $+0.16 \pm 0.07$ Wm$^{-2}$, while the corresponding ERF is $+0.35 \pm 2.1$ Wm$^{-2}$. This highlights the importance of accurately representing the volatility of such compounds in global scale models to improve the accuracy of capturing aerosol properties and their impacts on the climate. Furthermore, our comparison between the highly
resolved VBS setup with 9 volatility bins and a simpler VBS setup with 3 volatility bins revealed a remarkable difference in

N100 and CDNC. The 3-bin VBS setup exhibited higher N100 and CDNC compared to the 9-bin VBS setup, but lower SOA mass concentrations. These findings highlight the need for careful assessment when reducing the number of volatility bins and selecting appropriate values for volatility. We applied the arithmetic mean to calculate volatility in the 3-bin VBS setup based on the 9-bin VBS setup. While using the geometric mean for calculating the volatility resulted in improved agreement in SOA mass between the two setups, it led to less accurate matching for N100. Hence, choosing a value for volatility is a balance between getting correct particle number concentration or SOA mass concentration.

For future studies, it would be valuable to investigate the optimal VBS setup for a small number of volatility bins, aiming to strike a balance between computational efficiency and scientific accuracy. Such research could provide insights into achieving optimal speed and accuracy in modelling efforts related to aerosol properties and their implications for the climate. Moving forward, our study also underscores the necessity of incorporating data from volatility experiments involving diverse compositions into global modelling studies. Currently, volatility observations mostly focus on single compositions, limiting their ability to capture atmospherically relevant conditions. Components like brown carbon, dust aerosols, and sea sprays are prevalent in the atmosphere and requires their inclusion in volatility-based studies. Integrating such data is crucial for constraining the parametrizations of VBS bins in global models, thereby ensuring a comprehensive representation of aerosols.

*Code and data availability.* The ECHAM6-HAMMOZ model is provided to the scientific community under the HAMMOZ Software License Agreement, which outlines the terms and conditions for its usage. The license document can be obtained from https://redmine.hammoz.ethz.ch/attachments/291/License_ECHAM-HAMMOZ_June2012.pdf. Model data can be replicated using ECHAM-HAMMOZ model revision 6726, available in the repository https://redmine.hammoz.ethz.ch/projects/hammoz/ (HAMMOZ consortium, 2012). All the ECHAM simulation setup files, dataset, and Python scripts for data analysis are available from https://doi.org/10.23728/fmi-b2share.6416b bff3bb24b3eb1d49cd990fda411 (Irfan et al., 2023). All emission input files are from the standard ECHAM-HAMMOZ and are accessible through the HAMMOZ repository (refer to https://redmine.hammoz.ethz.ch/projects/hammoz). The code for MCOLNAG model is available upon request from the corresponding author.

*Author contributions.* HK, TYJ, AV, TK and MI planned the study. MI performed all the climate model simulations and produced the figures. TYJ performed process model simulations and formulated the volatility distribution for the 9-bin VBS setup. HK and TK supervised the study. All authors contributed to the scientific discussion and interpretation of the results. MI wrote the manuscript with contributions from HK, TYJ, and AV, incorporating comments from all co-authors.

*Competing interests.* At least one of the (co-)authors is a member of the editorial board of Atmospheric Chemistry and Physics. The peer-review process was guided by an independent editor, and the authors also have no other competing interests to declare.

*Acknowledgements.* For computational resources, we acknowledge CSC – IT Center for Science, Finland. The ECHAM-HAMMOZ model

is developed by a consortium composed of the ETH Zürich, Max Planck Institut für Meteorologie, Forschungszentrum Jülich, University of Oxford, Finnish Meteorological Institute and Leibniz Institute for Tropospheric Research and is managed by the Center for Climate Systems Modeling (C2SM) at ETH Zürich.

*Financial support.* This research has been supported by the European Union's Horizon 2020 research and innovation programme under grant agreement no. 821205 (FORCeS), the Academy of Finland (grant no. 357905 and 317390), the University of Eastern Finland Doctoral

Program in Environmental Physics, Health and Biology, the European Research Council via project PyroTRACH (ERC-2016-COG) funded from H2020-EU.1.1. - Excellent Science (project ID 726165), and the Horizon Europe programme under Grant Agreement No 101137680 via project CERTAINTY (Cloud-aERosol inTeractions  their impActs IN The earth sYstem)..

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
