# Peer review of "A model study on investigating the sensitivity of aerosol forcing on the volatilities of semi-volatile organic compounds"

_EGUsphere, 2023_

## Author Comment (AC1)

We express our gratitude to both reviewers for their valuable feedback, which has significantly helped improve the quality of the manuscript. Our detailed responses to the reviewers' comments are provided below. The reviewers' comments are in **bold**, the authors' responses are in regular font, and the revisions are in purple.

**Reviewer 1:**

RC1: **This study underscores the critical role of accurately representing semi-volatile organics in global scale models for simulating aerosol-climate interactions. The volatility distribution of organic compounds is pivotal as it governs their phase partitioning. The authors explored the sensitivity of CCN to volatility assumptions of organic compounds. Findings indicate that while the process model shows high CCN sensitivity to volatility uncertainties, the global model, particularly with a detailed 9-bin VBS, reveals that particle number concentration and CDNC are largely unaffected by major shifts in volatility. Most of the mass in lower volatility bins remained in the particle phase. The SOA mass burden, N, and radiative forcing were found to be more responsive to uncertainties in semi-volatile bins than in low-volatile bins. Furthermore, the study identified that IRFari is sensitive to volatility uncertainties in a detailed 9-bin VBS setup, while the ERF becomes sensitive in a simplified 3-bin setup. Comparing the 9-bin and 3-bin VBS setups, the study found notable differences in N100, CDNC, and SOA mass concentrations. The 3-bin setup showed higher N100 and CDNC but lower SOA mass concentrations compared to the 9-bin setup. The choice of volatility calculation method also impacted the accuracy of the model, affecting either particle number concentration or SOA mass concentration.**
**Overall, the results from this manuscript are interesting; my main concerns are the description of the model and methods as well as the explanation of some of the results, which were hard to follow, as mentioned below. Those parts would benefit from reorganizing and clarifications.**

Thank you for your comprehensive review and insightful comments on our manuscript. We appreciate the time and effort you dedicated for providing a constructive feedback. In response to your comments, we have carefully revised the manuscript to address each of your concerns and enhance the clarity and readability of the content. Below is a detailed response to the specific comments you raised:

**60:"GISS II' GCM (Farina et al., 2010), ECHAM-SALSA (Mielonen et al., 2018), CESM2 (Tilmes et al., 2019), GEOS-CHEM (Fritz et al., 2022), GFDL AM (Zheng et al., 2023)" please spell out all the models.**

Thanks for pointing this out. We have spelled out the full names of the models in the revised manuscript as follows:
Given the wide spectrum of SOA precursor species, their concentrations, and compositions, several of the current climate models (for example, ECHAM-SALSA (European Centre Hamburg Model - Sectional Aerosol module for Large Scale Applications) (Mielonen et al., 2018), CESM2 (Community Earth System Model 2) (Tilmes et al., 2019), GEOS-CHEM (Goddard Earth Observing System - Atmospheric Chemistry) (Fritz et al., 2022), GFDL AM (Geophysical Fluid Dynamics Laboratory's Atmosphere Model) (Zheng et al., 2023) use volatility basis set (VBS) approach to represent SOA and to simulate the formation of SOA in the atmosphere.

**60: Citing Farina et al (2010)as GISS II GCM needs double checking, according to Farina, "The primary tool in this work is the so-called "unified" chemistry-climate-aerosol model [Liao and Seinfeld, 2005], which was developed on the basis of the Goddard Institute for Space Studies General Circulation Model II (GISS II GCM) [Rind and Lerner, 1996] and includes online tropospheric chemistry [Mickley et al., 1999; Wild et al., 2000] and bulk aerosol**

thermodynamics modules [Adams et al., 1999; Nenes et al., 1999]. " it was based on GISS II GCM with other configurations to make it, as Farina et al called it, "Global Chemical Transport Modeling" , please differentiate with GISS ModelE, which is currently more well acknowledged in the modeling community.

We appreciate the feedback regarding the citation of Farina et al. (2010) and the distinction between GISS II GCM and GISS ModelE. In the revised manuscript, we have removed the citation of Farina et al. (2010) to avoid any potential confusion.

**64-66: please cite Donahue et al. (2006) as well since it was the first time the original concept was presented**

We appreciate the suggestion to include the citation of Donahue et al. (2006) as it was the first presentation of the original concept. We have now included this citation in the revised manuscript.

**66-69: this is a very long complex sentence, please consider re-writing to shorter ones for easier understanding and readability**

We have revised the sentence to address your concern about sentence length and complexity. The sentence was modified as follows:

The VBS framework aims to simplify the complex and often poorly understood processes involved in SOA formation. This simplification is achieved by combining compounds to groups based on their volatility reducing the complexity of chemical processes involved in the formation and aging of SOA (Donahue et al., 2006).

**83: does SALSA have a full name? please spell it out**

SALSA is acronym for Sectional Aerosol module for Large Scale Application. We have spelled out the full name of SALSA in the revised manuscript.

**98-107: Unfortunately, I had trouble following the description of the particle growth model even after reading it 4 times. Please, please re-write or include an illustration. Currently, the major confusions for me are :**

- **particle and nanoparticles need to be specific, in the first sentence they're both used. What size are we talking about?**

  The sentence was clarified to explicitly state that we model the growth of the nucleation mode particles. The sentence now reads as:

  In this study, we simulated the growth of nucleation mode particles and their survival to CCN size on process-scale with a Model for coagulation losses in nanoparticle growth (MCOLNAG).

- **the modes aren't specified either, please differentiate nucleation mode, Aitken mode, and accumulation mode in the text.**

  The modes have been specified in the revised manuscript. We have differentiated between nucleation mode, Aitken mode, and accumulation mode in the text in L124 as follows:

  Number concentrations were $1161\,\mathrm{cm}^{-3}$ and $321\,\mathrm{cm}^{-3}$ and diameters were $53\,\mathrm{nm}$ and $170\,\mathrm{nm}$ for Aitken and accumulation modes, respectively. Initial number concentration of nucleation mode was $539\,\mathrm{cm}^{-3}$ and initial diameter of the nucleation mode particles was $3\,\mathrm{nm}$.

- **"Particle phase is assumed to form an ideal solution and no particle phase processes are included." this is also confusing considering you mentioned particle properties, and**

**particle phase repeatedly, if particle phase processes aren't considered, how are properties changed and what's going on with coagulation, etc. in the next sentence?**

By particle phase processes, we referred to processes taking place internally in a particle, i.e. chemical reactions and diffusivity limitations inside the particle arising from viscosity. To avoid confusions, we have revised this sentence as follows:

Particle phase is assumed to form an ideal solution, behave liquid-like, and no particle phase chemical reactions are included.

- **"Aitken and accumulation mode particle diameters and number concentrations are set constant. Self-coagulation decreases nucleation mode number concentration but its impact to particle size is ignored." But why? You're talking about a model that simulates "particle growth and survival to CCN size" How do you set diameters to constant and ignore self-coagulation's impact on size?**

The model focuses on modelling the growth of the nucleation mode particles (see also the revision for the first sentence of the paragraph). Therefore, diameters of Aitken and accumulation modes are set constant. Impact of self-coagulation on nucleation mode particle size is not accounted for in the model as it is assumed small compared to the effect of condensation growth in the simulated conditions. L118 was revised as follows:

Self-coagulation decreases nucleation mode number concentration but its impact to particle size is ignored, i.e. the nucleation mode particles grow in the model by condensation only.

**Perhaps re-writing this description more clearly would help clear the confusion.**

We have revised the section to ensure that it is presented in a clearer and more understandable manner.

**109-110: "stoichiometric coefficient of the respective bin." Can you list them? If they're listed later, please refer to them here too**

We appreciate your suggestion to refer them explicitly. The stoichiometric coefficients are listed in the later section in the manuscript. In the revised manuscript, we refer to section 2.3 in sentence as follows:

Concentrations of each bin were defined by setting the total concentration of organic vapors and multiplying that with the stoichiometric coefficient of the respective bin (see Sect. 2.3 for stoichiometric coefficients).

**117-134: let's walk through the logic here: model+microphysics module SALSA is mentioned, model description, then aerosol module HAM description, then HAM's two options M7 & SALSA, then SALSA in more detail. It does not flow well. Please consider reorganizing the order, for example, from big to small: main model -> aerosol module -> aerosol microphysics module options.**

We understand the concern regarding the organization of model descriptions and agree that a more logical flow would improve readability. In the revised manuscript, we reorganized and modified the content in section 2.2 to follow a hierarchical structure, starting from the main model, then moving to the aerosol microphysics module SALSA.

**135-149: I see there are more descriptions for SALSA, which is good. I recommend going through model descriptions (2.1-2.2) as a whole section and seeing the logic and flow to make it clearer, and better connection with the whole model. It's very unclear at the moment how they're connected. They're just paragraphs of information with little connections. For**

instance, reading up to this point, it's unclear how MCOLNAG plays a role in the global model and its relations to SALSA.

Thank you for highlighting the need for better clarity in the model descriptions. In the context of the particle growth model, the relationships between MCOLNAG, SALSA, and the global model need clarification. Here, we highlight that the MCOLNAG model is used to simulate particle growth and losses at the process scale. It specifically focuses on uncertainties in volatility and their impact on Cloud Condensation Nuclei (CCN) sized particles. Importantly, MCOLNAG works independently of SALSA and the global model. SALSA, on the other hand, captures aerosol processes at a broader scale which includes several aerosol microphysics. The two models, MCOLNAG and SALSA, are distinct entities in the study, each serving a specific purpose within their respective scales. The global model, incorporating SALSA, is then used to study how uncertainties observed at the process scale, as simulated by MCOLNAG, translate to the larger context. It is crucial to emphasize that, in this study, the process-scale model (MCOLNAG) and the global model (ECHAM-SALSA) are not directly related but serve as complementary tools for investigating different aspects of aerosol dynamics, with the former focusing on process-scale uncertainties and the latter on global-scale implications. This clear distinction helps in understanding the study's structured approach, starting from process-level and extending them to global implications. We have modified the section 2.2 in the revised manuscript to improve the overall flow of information.

**205: "This is why the VBS setup of several global models have simpler VBS representations." Please specify which ones**

Here, we meant that global models generally use fewer volatility bins. For instance, WRF-CHEM (Reyes-Villegas et al., 2022), CESM2 (Tilmes et al., 2019), and the previous version of ECHAM-SALSA (Mielonen et al., 2018) have VBS setups with only 3 to 4 volatility bins. We have included this information in the revised manuscript in L226 as follows:

This is why the VBS setup of several global models, for instance, WRF-CHEM (Reyes-Villegas et al., 2022), CESM2 (Tilmes et al., 2019), and the previous version of ECHAM-SALSA (Mielonen et al., 2018) have simpler VBS representations.

**301-306: Fig 5 shows that there's a shift in mass, but barely changed for N and CDNC. Could you please explain why this is? This paragraph only described the results but there's no discussion of the reason why.**

Thanks for your comment and observation regarding the findings presented in Fig. 5. In our study, we found that across all VBS setups, low volatility compounds, which play a crucial role in the growth of smallest particles to CCN sizes, tend to partition into the particle phase. This partitioning influences the growth of particles into sizes relevant for cloud formation. Notably, the most substantial difference in partitioning was observed for VBS bins with higher volatilities. These compounds, with their higher volatility, exert a greater influence on particle mass with larger particles that are already within the range of CCN sizes. Hence, while the overall mass may shift, the CCN concentration and CDNC may remain relatively unchanged due to the complex interplay of particle growth mechanisms and the distribution of volatile compounds. We have included the following discussion in the revised manuscript:

The observed shifts in SOA mass concentration and particle number concentration could be attributed to the partitioning behaviour of volatile compounds within the aerosol population. Specifically, low volatility compounds, which play a significant role in the growth of smallest particles to CCN sizes, tend to partition to the particle phase across all VBS setups as seen in Fig. 3. Notably, the biggest differences in partitioning occur for VBS bins with higher volatilities, which have more influence on particle mass in larger particles already in CCN sizes. Hence, while overall mass changes with volatility shifts, the CCN concentration and CDNC may remain relatively unchanged.

**310: This is interesting, but which factors? Could you please elaborate?**

Here, we wanted to highlight that the change observed in CCN concentration in response to variations in aerosol volatility does not correspondingly influence CDNC. The discrepancy suggests that while CCN experiences a discernible shift due to changes in volatility, the subsequent formation and concentration of cloud droplets remain relatively unaffected. This could be attributed to the interplay of multiple factors influencing CDNC, such as cloud microphysics, meteorological conditions, and the aerosol composition. We elaborate this in the revised manuscript as:

This could be due to the fact that changes in CDNC are influenced by a combination of factors beyond CCN concentration alone, including cloud microphysics, meteorological conditions, and the aerosol composition

**319-323: "The difference in mass concentration primarily comes from the third bin (bin3) of the 3-bin VBS setup, which contributes to the lower SOA mass concentration in the 3-bin VBS setup, while the mass concentration in the other two lower volatile bins is the same between the two VBS setups. Differences between the two setups are sensitive to how the volatilities for each bin have been chosen in the 3-bin VBS setup."**

- **It's hard to follow what this is trying to say. Which is the third bin, it is not clear in Fig 6. Which are the two lower volatility bins and which are you comparing against, the a vs b and c?**

  Thank you for your comment. In our analysis, we are comparing the difference in mass concentration between the 9-bin VBS setup and the 3-bin VBS setup. Here, we refer to the difference in mass concentration between the two VBS setups, which primarily arises from the differences observed in the higher volatile bins between these two setups. Specifically, the mass concentration from bin3 (third bin) in the 3-bin VBS setup and the total mass concentration from the corresponding volatility bins from 9-bin VBS: $10^1$, $10^2$, and $10^3$ exhibit notable variations. It is crucial to note that while there are notable differences in the mass concentration in higher volatile classes, the mass concentration in bin1 and bin2 in the 3-bin VBS setup closely aligns with the total mass concentration in their counterparts in the 9-bin VBS setup as shown in Fig. 1. To depict this difference clearer, we include a figure in the supplementary materials and refer to section 2.3, where the bins are described. This comparison in mass concentration between two VBS setups underscores the sensitivity of our model outcomes to the choice of volatility bins, particularly in the higher volatile classes, highlighting the importance of accurately defining the volatility distribution for robust aerosol simulations. We have elaborated this paragraph in the revised manuscript with the following text:

  The difference in mass concentration between the two VBS setups primarily comes from the variations in mass concentration within the bin3 (third bin) of the 3-bin VBS setup compared to the cumulative mass concentration in the corresponding bins of the 9-bin VBS setup (See Supplementary Material Fig. S2). This results in a lower SOA mass concentration observed in the 3-bin VBS setup. However, It's worth noting that the mass concentrations in the other two lower volatile bins in the 3-bin VBS setup remains consistent with the cumulative mass concentrations in their corresponding bins from 9-bin VBS setup. This differences between the two setups highlights the sensitivity of their configurations to the selection of volatilities assigned to each bin in the 3-bin VBS setup.

- **Similar problem for Figure 7. The referred bins are hard to follow. I recommend that you add a summary table of all the experiments for better reference in the text.**

  In response to your question, we have addressed the issue by adding a table that describes the names of the bins in the 3-bin VBS setup alongside their corresponding $C_{sat}$ ranges and their equivalents in the 9-bin VBS setup as shown in Table **??**. We also include a summary of experiments with all the simulations used in this study as shown in Table 2. These tables offer a clear understanding on the naming conventions used in the study along with all the simulations, facilitating a better understanding

[Figure]

Figure 1: Mean vertical profile of VBS only mass concentrations from **(a)** bin3 **(b)** bin2 and **(c)** bin1 in 3-bin VBS setup and the combined mass concentrations in their corresponding bins from 9-bin VBS setup. The error bar depicts one-sigma standard deviation of the data calculated across different grid cells over each model levels for the boreal forest region.

of how the volatility bins are represented in each configuration. We have also included theses tables in the manuscript to help comprehending the differences between the two VBS setups.

Table 1: Overview of Saturation Concentrations and Stoichiometric Coefficients used for 9-Bin and 3-Bin VBS setups

| 9-Bin VBS | | 3-Bin VBS | | |
|---|---|---|---|---|
| $C_{sat}$ | $\alpha$ | $C_{sat}$ | $\alpha$ | Bin |
| 0 | 0.0038 | | | |
| $10^{-4}$ | 0.0029 | $4.8 \times 10^{-4}$ | 0.0126 | Bin 1 |
| $10^{-3}$ | 0.0059 | | | |
| $10^{-2}$ | 0.0146 | | | |
| $10^{-1}$ | 0.016 | $5.48 \times 10^{-1}$ | 0.0639 | Bin 2 |
| $10^{0}$ | 0.0333 | | | |
| $10^{1}$ | 0.1028 | | | |
| $10^{2}$ | 0.1456 | $5.32 \times 10^{2}$ | 0.497 | Bin 3 |
| $10^{3}$ | 0.2491 | | | |

**Bonus question: do you think your results would hold true in other models, why and why not (and what factors would make the difference)?**

The main factor would be how the partitioning of VBS species is calculated in models. In our model, partitioning is calculated solving the condensation equations, However, some models do not account for the non-equilibrium growth of SOA (e.g., Zhang et al. (2012)). Instead they calculate equilibrium for SOA species, thus missing the growth of freshly formed particles to CCN sizes. In some cases, the growth of freshly formed particles is "prescribed" in the sense that the formed SOA precursor mass is distributed based on surface area of different sized particles after which evaporation is neglected (e.g., Bergman et al. (2021)). Such differences would result in differences in the response of CCN concentrations and SOA mass to SOA volatility.

It is also important to note that the degree of sensitivity observed in our study may be model-specific, depending on the underlying assumptions, parameterizations, and representations of aerosol processes in

Table 2: Summary of Experiments

| Experiment | VBS Setup | Volatility Shift | Experiment Description |
|:---:|:---:|:---:|:---:|
| 1 | 9-bin | VBS×1 | Original volatility |
| 2 | 9-bin | VBS×10 | Increased volatility by one order of magnitude |
| 3 | 9-bin | VBS×0.1 | Decreased volatility by one order of magnitude |
| 4 | 3-bin | bin1×10 | Increased volatility of bin1 by one order of magnitude |
| 5 | 3-bin | bin1×0.1 | Decreased volatility of bin1 by one order of magnitude |
| 6 | 3-bin | bin2×10 | Increased volatility of bin2 by one order of magnitude |
| 7 | 3-bin | bin2×0.1 | Decreased volatility of bin2 by one order of magnitude |
| 8 | 3-bin | bin3×10 | Increased volatility of bin3 by one order of magnitude |
| 9 | 3-bin | bin3×0.1 | Decreased volatility of bin3 by one order of magnitude |

different models. Aerosol model's representation—whether it adopts a sectional or modal approach is also a crucial factor (see e.g. Kokkola et al. (2014)).

Overall, we assume that, while our study provides valuable insights into the impacts of volatility uncertainties on aerosol and climate properties, the applicability of these results to other models requires careful understanding of the specific model configurations and processes employed in those models. Factors such as the complexity of the aerosol representation, treatment of semi-volatile compounds, and choices in the volatility basis set setup will play a significant role in determining the consistency of our findings across different modeling frameworks. Future collaborative efforts and model inter-comparison studies will contribute to a more comprehensive understanding of aerosol-climate interactions across diverse modeling frameworks.

**Reviewer 2:**

**General comment**

RC1: **This manuscript explores the sensitivity of aerosol growth rates (in a box model) as well as SOA mass, CCN, and radiative forcing (in a global model) to uncertainties in SOA volatility distributions. The analysis generally seemed sound, though the overall science contribution is generally minor (e.g., tying the box and global model results together to learn more about *why* SOA mass and CCN change in opposite directions could have made the paper a larger contribution). I'm fine with the manuscript being published once the authors have responded to my comments.**

Thank you for your insightful comments and feedback regarding the sensitivity analysis carried out in our manuscript. We appreciate your feedback on the soundness of our analysis. Regarding your point about the potential for a larger contribution of this manuscript by providing why SOA mass and CCN change in opposite directions, we agree that better elucidating the underlying processes driving these changes could improve the significance of our finding. However, we respectfully disagree with the point that SOA mass and CCN change in opposite directions in our results. While the process model shows a more pronounced change in SOA mass and the CCN, the global model also exhibits changes in the same directions, but to a lesser extent. This difference in the magnitude of change between the two models can be attributed to the inherent differences in their spatial and temporal scale as well as the inclusion of more atmospheric processes including removal mechanisms in the global model. The process model, with its focus only on microphysical processes like condensation and coagulation, tends to exhibit higher sensitivity to changes in SOA precursor volatility due to its simplified representation. On the hand hand, the global model provided a broader perspective by incorporating a wider range of atmospheric processes, including deposition and transport, which could affect the overall sensitivity observed in the process scale. We have revised the manuscript to

clarify this point and have addressed your comments comprehensively.

- **The box model and global model results are not tied well together. They feel like completely orthogonal analyses in the manuscript. What can we learn from using both tools? What box-model simulations can help elucidate some of the findings of the global model?**

  Thanks for your comment. One motivation for this study was to examine how process model results, which only take into account microphysical processes, translate to global scale, which also includes several processes that can buffer the changes SOA makes to the aerosol population. The process model, with its focus only on microphysical processes such as condensation and coagulation, provides a detailed understanding of aerosol behaviours at a fine-grid process scale. On the other hand, the global model offers a more broader and realistic perspective, allowing us to explore aerosol dynamics on a larger spatial and temporal scale. With all the feedback processes and complexities incorporated into the global model, we ensured to simulate aerosol behaviour at an atmospherically relevant scale, enabling us to capture a more comprehensive picture of aerosol dynamics. We have modified the manuscript in L85 to improve the linkage between the two models to provide a better narrative as given below:

  In this study, we used a process-scale growth model and a global aerosol-climate model to investigate the sensitivity of volatility distribution to SOA formation and cloud properties. The process model focused on examining the sensitivity of particle growth rate and their survival across CCN size ranges to uncertainties in the volatilities of organic vapours in a process scale. One motivation for this study was to examine how process model results, which only take into account microphysical processes, translate to global scale, which also includes several processes than can buffer the changes SOA makes to the aerosol population. To understand how these sensitivities in the process scale manifest at a global scale, we also utilized the global aerosol-climate model ECHAM-HAMMOZ coupled with aerosol microphysical model SALSA, which is the Sectional Aerosol module for Large Scale Applications (Kokkola et al., 2018; Holopainen et al., 2020, 2022).

- **If I were to choose the C\* (Csat) of the 3 volatility bins to best capture both particle growth (CCN) and SOA mass, I'd choose: one low volatility bin that condenses irreversibly and contributes to growth of small particles (Csat of 1E-2 ug m-3 and lower grouped together) and two bins that span the range of atmospheric OA concentrations (maybe Csat of 0.5 and 5 ug m-3). Unless the model includes aging through the VBS bins, the 5.32E2 ug m-3 bin is going to be irrelevant for SOA formation during warmer months when monoterpenes are being emitted. I was surprised to not see any reasoning through what an ideal 3-bin scheme might be to capture both growth and mass variability. The readers are not left with a way forward for SOA modeling.**

  Thanks for the insightful comment and suggestion. We agree that choosing appropriate $C_{csat}$ values is crucial for the accurate representation of both particle growth and SOA mass. While we aimed to provide a comprehensive discussion on the rationale behind selecting the $C_{csat}$ values for the volatility bins, we acknowledge that we overlooked a more detailed explanation of an ideal 3-bin VBS setup. The proposed setup, with one low volatility bin and two bins spanning atmospheric OC concentrations seems reasonable, and could improve the modelling of SOA formation. However, from Fig. 3 in the manuscript, it's quite evident that the gas-particle partitioning in the lower volatility bins ranging from $C_{csat}$ values 0 to $10^{-1}$ is not sensitive to their volatilities, while the $C_{csat}$ bins ranging from $10^0$ to $10^3$ seems to have a notable sensitivity to their volatilities. This implies that, as mentioned, all bins from $C_{csat}$ values 0 to $10^{-1}$ can be grouped together to one single bin and thereby improving the resolution of other semi-volatile organics with two bins spanning atmospheric OA concentration.

We have revised the conclusion section in the manuscript to include a detailed discussion to provide a better understanding on the selection of volatility of VBS bins and the way forward in SOA modelling.

The revised text in L442 is as follows:

This behavior of gas-particle partitioning indicates that the lower volatility bins are insensitive to reasonable uncertainties in their volatilities, whereas the higher volatilities with $C_{sat}$ values of $10^{-1}$ $\mu g/m^3$ or higher are highly sensitive to their volatilities. This suggests that the semi-volatile bins in the VBS setups require higher resolution than the low volatile compounds in a global model. Essentially, if a global model needs to represent organic compounds in fewer volatility bins, then ideally, the lower volatile bins with $C_{sat}$ values up to $10^{-1}$ can be lumped together into a single bin, while the remaining bins can have better representation.

The revised text in L464 is as follows:

Moving forward, our study also underscores the necessity of incorporating data from volatility experiments involving diverse compositions into global modelling studies. Currently, volatility observations mostly focus on single compositions, limiting their ability to capture atmospherically relevant conditions. Components like purple carbon, dust aerosols, and sea sprays are prevalent in the atmosphere and requires their inclusion in volatility-based studies. Integrating such data is crucial for constraining the parametrizations of VBS bins in global models, thereby ensuring a comprehensive representation of aerosols.

**Specific Comments**

**L26-28: These values are very large. When I read the abstract, I thought these were global forcings, but I think they are just over the boreal forest regions, so this needs to be clear here.**

Thank you for your comment. The radiative forcings mentioned pertain specifically to the boreal forest regions and not global averages. The sentence now reads as:
In the 3-bin VBS setup, a ten-fold decrease in volatility of the highest volatility bin results in a shortwave instantaneous radiative forcing (IRF$_{ari}$) of -0.2 $\pm$ 0.10 Wm$^{-2}$ and an effective radiative forcing (ERF) of +0.8 $\pm$ 2.24 Wm$^{-2}$, while a ten-fold increase in volatility leads to an IRF$_{ari}$ of +0.05 $\pm$ 0.04 Wm$^{-2}$ and ERF of +0.45 $\pm$ 2.3 Wm$^{-2}$ over the boreal forest regions.

**L26-28: What do the uncertainty ranges here represent? Why are the uncertainty ranges for the ERFs so large (relative to the central value and the IRF uncertainty ranges)?**

The uncertainty ranges represent the spatial variability across all grid cells for the estimated radiative forcings. The wide uncertainty ranges in the radiative forcing estimates could be attributed to the limitations of having data for just one year of summer simulation. This relatively short duration may not fully capture the variability in climate conditions and feedback mechanisms, leading to larger uncertainty ranges in the estimated radiative forcings. One significant cause for the high uncertainty range is also the spatial variability of the simulated cloud properties.

**L32-33: The total atmospheric aerosol loading by mass is dominated by coarse aerosol: dust and sea spray, not organics. The papers cited for the range are for submicron aerosols, mostly at continental sites, so "total atmospheric aerosol loading" is not precise.**

Thanks you for pointing this out. We have modified this sentence to read:
These atmospheric aerosol make up 20-90% of the total submicron aerosol loading (Jimenez et al., 2009; Hallquist et al., 2009), and they are crucial for both human health and the climate.

**L43: I don't think the consensus is currently that aromatics are the main anthropogenic SOA precursors (S/IVOCs are in the mix now), though I guess you say "VOC" in the sentence.**

The referee is correctly pointing out the consensus has evolved regarding the major anthropogenic SOA precursors. Several studies have recognized that alkanes play a significant roles in SOA formation alongside aromatics. We revised the manuscript as follows:

Terpenes (e.g., $\alpha$-pinene and $\beta$-pinene) and isoprene are dominant sources of biogenic VOCs globally, while alkanes and aromatics (e.g., toluene and xylene) are the major anthropogenic VOCs (Ziemann and Atkinson, 2012).

**L67: Partitioning is a phase transition, not a reaction.**

This is correct that partitioning is a phase transition rather than a reaction. We have revised the sentence to read as follows:

This simplification is achieved by combining compounds to groups based on their volatility reducing the complexity of chemical processes involved in the formation and aging of SOA.

**L72: Dominate what?**

Here, we meant that monoterpene emissions are the primary or most significant contributors to the overall VOC emissions in boreal forested regions. The sentence now reads:

Monoterpenes dominate the VOC emissions in boreal forested areas, covering 29% of forested land areas, thereby making this region a significant source of biogenic SOA (Rinne et al., 2009; Kayes and Mallik, 2020).

**L108-110: So the vapor concentrations are held fixed? There are no feedbacks as condensation sinks evolve? No volatility-dependent growth-rate differences between smaller and larger particles? Is this realistic enough to gain useful insight?**

In the atmosphere, the organic vapor concentrations are affected by various other processes than condensation sink, such as emissions and chemical reactions. As such, accounting for condensation sink may not necessarily lead to more atmospherically relevant simulations. Here, our aim was to make the process model simulations on as simple level as possible and to reduce the number of tunable parameters, and therefore we have decided not to include factors that change the vapor concentrations.

**L130: I'm confused because one sentence says that it's a grid model, the next says that it's a spectral model.**

We have corrected this to
ECHAM-HAMMOZ consists of the atmospheric general circulation model ECHAM6, which solves the equations of motion and continuity for the atmosphere using the spectral method

**Section 2.2: Semi-volatile condensation/evaporation across size bins is a very stiff numerical system. Small particles will adjust very quickly to be in equillibrium with gas-phase semi-volatile species while accumulation/coarse-mode particles take orders-of-magnitude longer. How does the model handle this? Is there some sort of hybrid solver scheme or do you take very short timesteps to explicitly resolve the cond/evap kinetics of the smallest particles? The text says the APC scheme, but can you state a bit more?**

It is true that it is a very stiff numerical system and to avoid oscillatory behaviour in condensation, we solve the condensation equations using five time steps to solve the condensation over one atmospheric

model time step, with condensation solver time step length increasing logarithmically. We have added this explanation in the revised manuscript in L176 as follows:

To avoid any oscillatory behaviours in condensation, we solve the condensation equations using five time steps to solve the condensation over one atmospheric model time step, with condensation solver time step length increasing logarithmically.

**L165: Csat is stated as a mass concentrations elsewhere in the paper. Is "x" in eqn 2 actually a mole fraction or is it a mass fraction. The Donahue VBS usually uses mass fraction with mass-based concentrations, while Raoult's law would use mole fraction with partial pressures (proportional to molar concentrations rather than mass concentrations). Using mole fractions with mass concentrations would be an unusual hybrid.**

In these equations, concentrations are mole concentrations $(mol/m^3)$ converted from mass concentration based values and "x" actually is a mole fraction. This has been clarified in the modified manuscript at L185 as follows:

In these equations, concentrations are expressed as mole concentrations $(mol/m^3)$, which are derived from mass concentration based values.

**L188: This is a very low Hvap that came out of not-fully-constrained fits of smog-chamber experiments. It's inconsistent with Hvap values for SOA C\* values of interest, e.g., `https://doi.org/10.1021/es902497z` (Epstein et al., EST, 2010).**

We admit that the Hvap value we used from Farina et al. (2010) may appear low and not fully constrained. However, we would like to emphasize that our choice of Hvap value aligns with previous literature and is consistent with values commonly used in many other global modelling studies, for eg., Patoulias et al. (2018); Glotfelty et al. (2017); Pathak et al. (2007).

**Section 3.1: Did you explore the interplay between volatility and the relative amount of mass split between the growing nucleation mode and the pre-existing larger models, e.g., `https://doi.org/10.5194/acp-11-9019-2011` (Pierce et al., ACP, 2011)?**

No, we did not explore the interplay between volatility and the relative amount of mass split between the growing nucleation mode and pre-existing larger modes. Our process model does not account for condensation onto pre-existing particles. Also, assuming constant vapor concentrations would "disconnect" the condensation growth of the nucleation mode and pre-existing particles. This means that even if the condensation growth of the Aitken and accumulation mode were included, their consumption of vapors would not affect the growth of the nucleation mode.

**Figures 2 and 4: While it does say at line 231 in the methods that the results will focus on the boreal regions, most readers skim articles and don't read methods in detail. Please add a note to the caption says that the figure is limited to boreal regions to highlight the sensitivity to monoterpene SOA (or similar).**

Thank you for pointing out this. We have included a note in the caption of Figures 2 and 4 to clarify that the results are specifically focused on boreal regions. The modified caption of Fig.2 now reads as:

Simulated SOA burden of **(a)** VBS×1 and the relative difference between **(b)** VBS×10 and **(c)** VBS×0.1 with respect to VBS×1, focusing specifically on boreal forest regions to emphasize the sensitivity to monoterpene SOA.

**Figures 3, 5, 6, 7, 8, and 9: Are these limited to just the regions above boreal forests (e.g. the regions in the maps of figures 2 and 4). Please state this explicitly in the captions. This**

information is critical to understanding these figures. Note that Figures 8 and 9 says "boreal region", but I think it's just the boreal forest region (if it is a more general "boreal" northern region beyond just the forests, then the region needs to be defined). Also, what does "close to ground level" mean specifically?

Thanks for the comment. Figures 3, 5, 6, 7, 8, and 9 focus exclusively on the boreal forests, as delineated in the maps of Figures 2 and 4. For clarity, we have modified the captions in the revised manuscript that these figures are limited to the boreal forested regions. Additionally, by "close to ground level", we mean the lowest vertical level in the model, representing lower atmosphere.

**Figure 3: I'm a bit confused about how there is non-trivial mass in the particle phase in the 1E3 ug m-3 bin in the VBSx10 simulation (which would make this bin 1E4 ug m-3, right?). It would need to be very cold or very high concentrations in order to get much aerosol mass in that bin. But I wouldn't expect much SOA precursor emissions when it's very cold, so in a weighted average, I'd expect very little mass in the particle phase in this bin. Can you please check the numbers in this figure?**

Thank you for bringing up this point. Firstly, we would like to clarify that the analysis presented in our manuscript focuses specifically on summer months. This is because of higher emission of monoterpenes during the warmer months, which is explained in the manuscript in L252. Since the partitioning of VBS compounds depends on the mole fraction, which also includes water, at high relative humidities the mole fraction value becomes low and also higher volatility compounds are partitioned to particle phase.

**Figures 5, 6, and 7: Are the concentrations on the x axis for ambient conditions or normalized to STP (e.g. ug sm-3)? This needs to be specified because it makes about a factor-of-5 difference in the concentrations at 200 hPa.**

The concentrations on the x-axis for Figures 5, 6, and 7 represent ambient conditions, not normalized to STP. To ensure clarity, we'll explicitly specify in the captions that the concentrations on the x-axis are in ambient conditions, not normalized to STP. .

**Figure 7: Too many colors are similar here, especially the reds, and I can't tell what is what.**

Thank you for your feedback. We have modified Figure 7 in the revised manuscript to ensure better differentiation, particularly among the similar red shades. The modified figure in the manuscript is added here as Fig. 2

**Figures 8 and 9 and the associated discussion: What are the total IRF and ERF of SOA in the base case (i.e., base relative to a case with SOA shut off)? This will give more context to the sensitivity of the forcings to SOA volatility here.**

We have performed an additional simulation with the SOA shut off. In the base case scenario (VBSx1) relative to a case with SOA shut off, the total instantaneous radiative forcing ($\text{IRF}_{ari}$) of SOA over the boreal forested region is found to be -0.18 $\text{Wm}^{-2}$, and the effective radiative forcing (ERF) is -1.03 $\text{Wm}^{-2}$. We have added this information in the revised manuscript in L409 as follows:

Additionally, we estimated $\text{IRF}_{ari}$ and ERF from a scenario with the SOA shut off relative to the VBS×1 simulation. The $\text{IRF}_{ari}$ of SOA over the boreal forested region is -0.18 $\text{Wm}^{-2}$, and the ERF is -1.03 $\text{Wm}^{-2}$, highlighting the significant cooling effect of SOA in this region.

**Figures 8 and 9: Error bar depicts 1 standard deviation of what? The different grid cells over boreal forests? Is it still temporally averaged or is temporal variability also part of the**

[Figure]

Figure 2: Mean vertical profile of **(a)** SOA mass concentrations and **(b)** N100 at ambient conditions when the VBS bins are shifted individually using the 3-bin VBS setup over the boreal forest region. The error bar depicts one-sigma standard deviation of the data calculated across different grid cells over each model levels.

**standard deviations?**

The error bars in Figures 8 and 9 represent 1 standard deviation of the data calculated across different grid cells over the boreal forests. This standard deviation shows the variability among the grid cells within the boreal forest regions. The data is temporally averaged over the summer season, meaning that the standard deviation includes the variability among grid cells over time within the averaged period. Hence, temporal variability is not explicitly represented in the standard deviations; rather, they include the spatial variability among grid cells during the averaged time period. We have clarified this in the revised manuscript at L268 as follows:

Additionally, we calculated one-sigma standard deviation across different grid cells over the boreal forests, indicating the variability within the boreal forest regions.

**Section 3.2.4: To increase the precision of writing, please try to always state what the IRFs and ERFs are relative to (in most cases in the discussion, it's the base simulation).**

Thanks for bringing this to our attention. We have made necessary revisions to explicitly state what the IRFs and ERFs are relative to.

**L405-406: I disagree that the ERF is only sensitive when the 3-bin version is used. Looking at Figure 8b, the ERF for VBSx10 and VBSx0.1 are at least as large as all of the IRF values in Figure 8a.**

It is true that that the ERF values for VBS×0.1 and VBS×10 in Figure 8b are of the same magnitude as the corresponding $IRF_{ari}$ values in Figure 8a. However, the uncertainty associated with the ERF changes is much bigger: for VBSx10, for instance, the IRF is $+0.16 \pm 0.07$ $Wm^{-2}$, while the corresponding ERF is $+0.35 \pm 2.1$ $Wm^{-2}$. In other words, the standard deviation for the $IRF_{ari}$ values is less than the change in the values, while for ERF, the standard deviation is about 6 to 7 times larger than the change. We therefore

consider the changes in ERF to be much less certain and therefore state that the ERF is only sensitive to the amount of bins used, not to changes in volatilities using the same amount of bins. We have added this clarification in the revised manuscript in L450 as follows:

The simulations also show that the $IRF_{ari}$ is sensitive to uncertainties in volatility in the 9-bin VBS setup while, the ERF becomes sensitive only if the simplified 3-bin VBS setup is used, due to the larger uncertainty associated with ERF changes. For example, in the VBS×10 scenario, the $IRF_{ari}$ is +0.16 ± 0.07 Wm$^{-2}$, while the corresponding ERF is +0.35 ± 2.1 Wm$^{-2}$.

**References**

Bergman, T., Makkonen, R., Schrödner, R., Swietlicki, E., Phillips, V. T., Le Sager, P., and Van Noije, T.: Description and Evaluation of a Secondary Organic Aerosol and New Particle Formation Scheme within TM5-MP v1. 1, Geoscientific Model Development Discussions, 2021, 1–43, 2021.

[revised manuscript text omitted]